# Improving Long-Text Alignment for Text-to-Image Diffusion Models

**Luping Liu**[*,1,2]**, Chao Du**[†,2]**, Tianyu Pang**[2]**, Zehan Wang**[*,2,4]**, Chongxuan Li**[3,5]**, Dong Xu**[†,1]

[1]The University of Hong Kong; [2]Sea AI Lab, Singapore; [3]Renmin University of China;

[4]Zhejiang University; [5]Beijing Key Laboratory of Big Data Management and Analysis Methods

`luping.liu@connect.hku.hk; duchao@sea.com; tianyupang@sea.com;`
`wangzehan01@zju.edu.cn; chongxuanli@ruc.edu.cn; dongxu@hku.hk`

## Abstract

The rapid advancement of text-to-image (T2I) diffusion models has enabled them to generate unprecedented results from given texts. However, as text inputs become longer, existing encoding methods like CLIP face limitations, and aligning the generated images with long texts becomes challenging. To tackle these issues, we propose *LongAlign*, which includes a segment-level encoding method for processing long texts and a decomposed preference optimization method for effective alignment training. For segment-level encoding, long texts are divided into multiple segments and processed separately. This method overcomes the maximum input length limits of pretrained encoding models. For preference optimization, we provide decomposed CLIP-based preference models to fine-tune diffusion models. Specifically, to utilize CLIP-based preference models for T2I alignment, we delve into their scoring mechanisms and find that the preference scores can be decomposed into two components: a text-relevant part that measures T2I alignment and a text-irrelevant part that assesses other visual aspects of human preference. Additionally, we find that the text-irrelevant part contributes to a common overfitting problem during fine-tuning. To address this, we propose a reweighting strategy that assigns different weights to these two components, thereby reducing overfitting and enhancing alignment. After fine-tuning $512 \times 512$ Stable Diffusion (SD) v1.5 for about 20 hours using our method, the fine-tuned SD outperforms stronger foundation models in T2I alignment, such as PixArt-$\alpha$ and Kandinsky v2.2. The code is available at https://github.com/luping-liu/LongAlign.

## 1 Introduction

Recent advancements in diffusion models (Sohl-Dickstein et al., 2015; Ho et al., 2020; Song et al., 2021a;b) have significantly enhanced text-to-image (T2I) generation (Schuhmann et al., 2022; Rombach et al., 2022; Saharia et al., 2022; Ramesh et al., 2022). While text-based conditioning provides flexibility and user-friendliness, current models struggle with long and complex text descriptions that often span multiple sentences and hundreds of tokens (Chen et al., 2023a; Zheng et al., 2024). Effectively encoding such lengthy text conditions and ensuring precise alignment between text and generated images remains a critical challenge for generative models.

To encode text descriptions, contrastive pretraining encoders such as CLIP (Radford et al., 2021) are widely used in T2I diffusion models. However, as text length increases, the maximum token limit of CLIP becomes a significant constraint, making it infeasible to encode long descriptions. Recent works have explored large language model (LLM)-based encoders like T5 (Raffel et al., 2020), leveraging their ability to handle longer sequences (Saharia et al., 2022; Chen et al., 2023a; Hu et al., 2024). Nevertheless, contrastive pretraining encoders retain a key advantage: their text encoders are specifically trained to align with images, potentially offering superior alignment between text representations and generated images (Saharia et al., 2022; Li et al., 2024b).

---

[*]Work done during Luping Liu and Zehan Wang's associate memberships at Sea AI Lab.
[†]Corresponding authors.

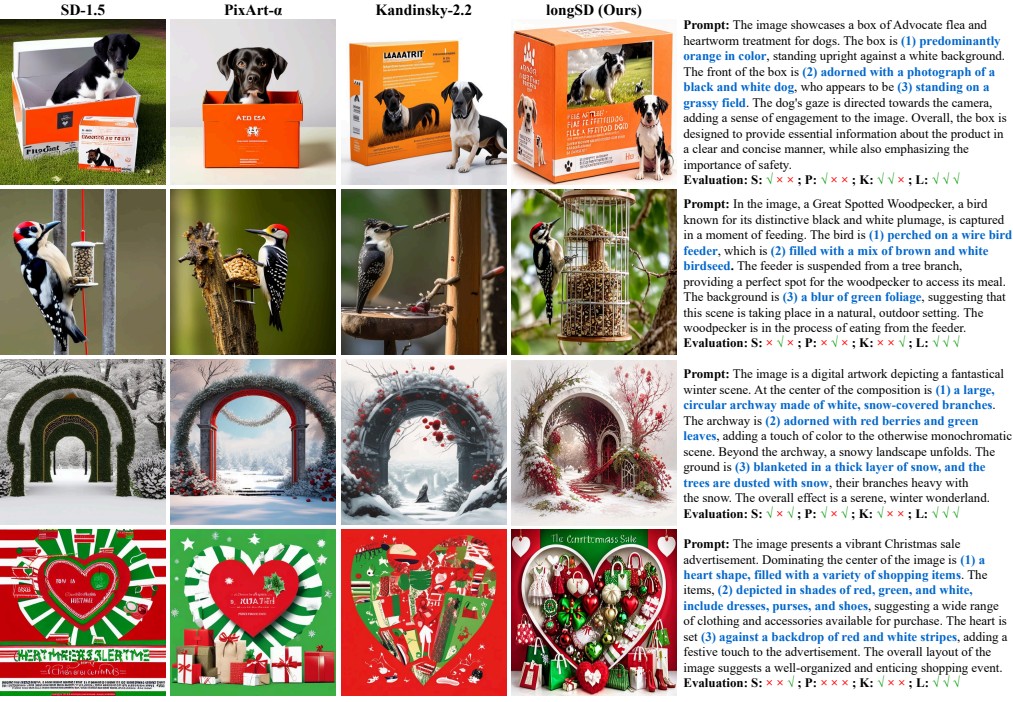

| SD-1.5 | PixArt-α | Kandinsky-2.2 | longSD (Ours) |
|---|---|---|---|

**Prompt:** The image showcases a box of Advocate flea and heartworm treatment for dogs. The box is **(1) predominantly orange in color**, standing upright against a white background. The front of the box is **(2) adorned with a photograph of a black and white dog**, who appears to be **(3) standing on a grassy field**. The dog's gaze is directed towards the camera, adding a sense of engagement to the image. Overall, the box is designed to provide essential information about the product in a clear and concise manner, while also emphasizing the importance of safety.
**Evaluation: S:** ✓×× ; **P:** ✓×× ; **K:** ✓✓× ; **L:** ✓✓✓

**Prompt:** In the image, a Great Spotted Woodpecker, a bird known for its distinctive black and white plumage, is captured in a moment of feeding. The bird is **(1) perched on a wire bird feeder**, which is **(2) filled with a mix of brown and white birdseed.** The feeder is suspended from a tree branch, providing a perfect spot for the woodpecker to access its meal. The background is **(3) a blur of green foliage**, suggesting that this scene is taking place in a natural, outdoor setting. The woodpecker is in the process of eating from the feeder.
**Evaluation: S:** × ✓× ; **P:** × ✓× ; **K:** ×× ✓ ; **L:** ✓✓✓

**Prompt:** The image is a digital artwork depicting a fantastical winter scene. At the center of the composition is **(1) a large, circular archway made of white, snow-covered branches.** The archway is **(2) adorned with red berries and green leaves**, adding a touch of color to the otherwise monochromatic scene. Beyond the archway, a snowy landscape unfolds. The ground is **(3) blanketed in a thick layer of snow, and the trees are dusted with snow**, their branches heavy with the snow. The overall effect is a serene, winter wonderland.
**Evaluation: S:** ✓× ✓ ; **P:** × ✓× ; **K:** ×× × ; **L:** ✓✓✓

**Prompt:** The image presents a vibrant Christmas sale advertisement. Dominating the center of the image is **(1) a heart shape, filled with a variety of shopping items**. The items, **(2) depicted in shades of red, green, and white, include dresses, purses, and shoes**, suggesting a wide range of clothing and accessories available for purchase. The heart is set **(3) against a backdrop of red and white stripes**, adding a festive touch to the advertisement. The overall layout of the image suggests a well-organized and enticing shopping event.
**Evaluation: S:** ×× ✓ ; **P:** ××× ; **K:** ✓×× ; **L:** ✓✓✓

Figure 1: Generation results of our long Stable Diffusion and baselines. We highlight three key facts for each prompt and provide the evaluation results at the end. In each evaluation line, the four group results are arranged in order of model presentation, with S representing SD-1.5, and so on. Additionally, three ✓ or × maintain the order of the key facts corresponding to each prompt.

Beyond encoding, current T2I diffusion models struggle to accurately follow long-text descriptions, often generating images that only partially reflect the intended details, as demonstrated in Figure 1. Inspired by advances in aligning LLMs (Ouyang et al., 2022; Rafailov et al., 2024), one potential solution is preference optimization, which generates and utilizes preference feedback when only parts of a target are satisfied. Recent works have explored collecting human preferences from T2I users and leveraging them to train preference models (Kirstain et al., 2023; Wu et al., 2023c;b), enabling preference optimization in T2I diffusion models. However, since these models are typically fine-tuned from CLIP, they face the same token limit constraints. Moreover, existing human preferences blend text alignment with visual factors such as photorealism or aesthetics, which only partially support the goal of accurately aligning long and detailed text descriptions.

In this paper, to support long-text inputs for the two scenarios mentioned, we explore segment-level encoding, which involves dividing lengthy texts into shorter segments (e.g., sentences), encoding each one separately, and then merging the results for subsequent tasks. The main challenge is effectively combining these segment outputs to merge their diverse information without causing confusion. To address this, for text encoding in diffusion models, we opt for concatenating segment embeddings and explore optimal adjustments to the unintended repetition of special token embeddings from different segments. For preference models, we implement a segment-level preference training loss alongside segment-level encoding, allowing preference models to handle long inputs while generating both segment-level scores and an overall average score.

To enable preference optimization for long-text alignment, we analyze the scoring mechanisms of preference models and use these models (with segment-level encoding) to fine-tune T2I diffusion models. We find that the desired T2I alignment scores can be separated from general human preference scores. Specifically, preference scores can be divided into two components: a text-relevant part that assesses T2I alignment and a text-irrelevant part that evaluates other factors (e.g., aesthetics). In addition, we discover that the remaining text-irrelevant part leads to a common overfitting issue (Wu et al., 2024) during fine-tuning. To mitigate this, we propose a reweighting strategy that assigns different weights to these two components, which reduces overfitting and enhances alignment.

By integrating the methods mentioned above, we propose *LongAlign*. Our experiments show that segment-level encoding and training enable preference models to effectively handle long-text inputs and generate segment-level scores. Additionally, our preference decomposition method allows these

models to produce T2I alignment scores alongside general preference scores. After fine-tuning the $512 \times 512$ Stable Diffusion v1.5 (Rombach et al., 2022) using LongAlign for about 20 hours on 6 A100 GPUs, the obtained long Stable Diffusion (longSD) significantly improves alignment (see Figure 1), outperforming stronger foundation models in long-text alignment, such as PixArt-$\alpha$ (Chen et al., 2023a) and Kandinsky v2.2 (Razzhigaev et al., 2023). Our contributions are as follows:

- We propose a segment-level encoding method that enables encoding models with limited input lengths to effectively process long-text inputs.

- We propose preference decomposition that enables preference models to produce T2I alignment scores alongside general preference, enhancing text alignment fine-tuning in generative models.

- After about 20 hours of fine-tuning, our longSD surpasses stronger foundation models in long-text alignment, demonstrating significant improvement potential beyond the model architecture.

## 2 BACKGROUND

We provide an overview of diffusion models and T2I models. Next, we discuss preference models for fine-tuning T2I diffusion models, followed by an introduction to the reward fine-tuning process.

### 2.1 DIFFUSION MODEL

Diffusion models (Sohl-Dickstein et al., 2015; Ho et al., 2020; Song et al., 2021a;b) construct a transformation from a Gaussian distribution to a target data distribution through a multi-step diffusion denoising process. Given a data distribution $x_0 \sim q(x_0)$, the diffusion process satisfies:

$$x_t = \alpha_t x_0 + \beta_t \epsilon, \tag{1}$$

where $\epsilon \sim \mathcal{N}(0,1)$, $t \in \{0, 1, \ldots, T\}$, $T$ is the maximum timestep, $\alpha_t^2 + \beta_t^2 = 1$, and $\beta_t$ controls the speed of adding noise. The loss function is typically defined as follows:

$$\mathcal{L}_t = \mathbb{E}_{x_0, \epsilon} ||\epsilon - \epsilon_\theta(\alpha_t x_0 + \beta_t \epsilon, t)||^2, \tag{2}$$

where the model $\epsilon_\theta$ aims to predict the noise $\epsilon$ added to the clean data $x_0$. Once $\epsilon_\theta$ is learned, according to DDIM (Song et al., 2021a), the denoising process from $t = T$ to $t = 0$ satisfies:

$$x_t^* = (x_t - \beta_t \epsilon_\theta(x_t, t))/\alpha_t, \quad x_{t-1} = \alpha_{t-1} x_t^* + \sqrt{\beta_{t-1}^2 - \sigma_t^2} \epsilon_\theta(x_t, t) + \sigma_t \epsilon, \tag{3}$$

where $x_T \sim \mathcal{N}(0,1)$ and $x_t^*$ is the predicted clean image at timestep $t$. When $\sigma_t \equiv 0$, this is the DDIM denoising process. When $\sigma_t = (\beta_{t-1}/\beta_t)\sqrt{(1 - \alpha_t^2/\alpha_{t-1}^2)}$, this is the DDPM denoising process. In this paper, we choose $\sigma_t \equiv 0$. The corresponding DDIM process can be accelerated using numerical solvers such as PNDM (Liu et al., 2022) and DPM-Solver (Lu et al., 2022).

**Stable Diffusion**. Among different T2I diffusion models, Stable Diffusion (Rombach et al., 2022) plays a crucial role, which integrates a VAE (Kingma & Welling, 2013), a CLIP, and a diffusion model $\epsilon_\theta$. During training, the pretrained VAE compresses the image $x$ into a latent $z$, while the pretrained CLIP model encodes the text prompt $p$. The diffusion model $\epsilon_\theta$ then learns to fit this new distribution of latent variables $z$, conditioned on the text $p$. In the sampling process, the diffusion model $\epsilon_\theta$ first generates the latent $z$ based on the text prompts $p$ and then uses the VAE to decode the latent $z$ to obtain the final image $x$. For simplicity, we will omit the VAE in the following.

### 2.2 PREFERENCE MODEL

Preference optimization (Ouyang et al., 2022; Rafailov et al., 2024) has shown its effectiveness in aligning LLMs with humans using preference feedback. To facilitate this for T2I diffusion models, prior work (Kirstain et al., 2023; Wu et al., 2023b) fine-tunes these models with T2I preference models that evaluate human preferences for an image $x$ given a text prompt $p$, represented as $\mathcal{R}(x, p)$. We focus on preference models fine-tuned from pretrained CLIP models. To prepare the preference dataset for fine-tuning, prompts are paired with two generated images, and the preferred image is annotated. For $x_i$ preferred over $x_j$ (denoted as $i \succ j$), the preference training loss is:

$$\mathcal{L}_{i \succ j} = \frac{\exp(\mathcal{R}(x_i, p))}{\exp(\mathcal{R}(x_i, p)) + \exp(\mathcal{R}(x_j, p))}, \tag{4}$$

where $\mathcal{R}(x, p) = \mathcal{C}_X(x) \cdot \mathcal{C}_P(p)$ is the dot product of the image embeddings $\mathcal{C}_X(x)$ and the text embeddings $\mathcal{C}_P(p)$. After training, the preference model can be used to evaluate preferences or fine-tune generative models. Similar to CLIP, these preference models don't support long-text inputs.

## 2.3 REWARD FINE-TUNING

When fine-tuning T2I diffusion models with the preference models mentioned above, previous works (Black et al., 2023; Fan et al., 2024) typically treat preference models as reward signals defined by $\mathcal{L}_r = 1 - \mathbb{E}_{x_T,p}\mathcal{R}(x_0^*, p)$. However, fine-tuning the generator $\epsilon_\theta$ of diffusion models using these signals poses two challenges. First, backpropagating gradients through the entire iteration process is problematic. Second, overfitting is a concern. Previous works (Prabhudesai et al., 2023; Clark et al., 2023) have employed gradient checkpointing and LoRA (Hu et al., 2021) to facilitate gradient backpropagation, but these methods skip consecutive time steps at the beginning or end of the diffusion iteration to accelerate computation, which inevitably introduces optimization bias.

Recently, DRTune (Wu et al., 2024) shows a new method that allows for training on a uniform subset of all sampling steps. Specifically, when we apply the preference model $\mathcal{R}(x_0^*, p)$ to calculate the reward signal $\mathcal{L}_r = 1 - \mathbb{E}_{x_T,p}\mathcal{R}(x_0^*, p)$, the corresponding gradient is as follows:

$$\partial_\theta(1 - \mathbb{E}_{x_T,p}\mathcal{R}(x_0^*, p)) \approx -\mathbb{E}_{x_T,p}(\partial_{x_0^*}\mathcal{R})(x_0^*, p)\left(\sum_{i=1}^{T-1}(\beta_{i-1}/\alpha_{i-1} - \beta_i/\alpha_i)(\partial_\theta\epsilon_\theta)(\text{sg}(x_i), i, p)\right). \quad (5)$$

DRTune truncates the gradient of all $\{\text{sg}(x_i)\}$, allowing it to optimize a uniform portion of the remaining gradient $\{(\partial_\theta\epsilon_\theta)(\text{sg}(x_i), i, p)\}$. In this paper, we also utilize this as our gradient backpropagation method. However, overfitting remains a concern that cannot be fully addressed by early stopping alone, necessitating further analysis and additional solutions.

## 3 SEGMENT-LEVEL TEXT ENCODING

In this section, we introduce our segment-level encoding method, designed for the long-text encoding of both diffusion models and preference models. This approach divides the text into segments, encodes each one individually, and then merges the results. For diffusion models, we explore an embedding concatenation strategy during merging. For preference models, we present a segment-level loss alongside the new encoding to handle long inputs and generate detailed preference scores.

### 3.1 TEXT ENCODING OF DIFFUSION MODEL

For text encoding in diffusion models, contrastive pretraining encoders like CLIP (Rombach et al., 2022) are commonly used. However, as the input text length increases, CLIP's maximum token limitation becomes a significant issue. As a result, recent works (Saharia et al., 2022; Chen et al., 2023a; Hu et al., 2024) have shifted to using LLMs like T5 instead of CLIP, overlooking CLIP's distinct advantage (Radford et al., 2021) in image-text alignment pretraining. To leverage CLIP's capabilities for long text, we introduce segment-level encoding. Our method divides the text into segments (e.g., sentences), encodes each one into embeddings like the original T2I diffusion models, and then merges these embeddings. Figure 9 illustrates this segment-level text encoding.

In the merging process, we initially use direct embedding concatenation, which results in poor generated images (see Figure 10). This occurs because each segment includes special tokens such as <sot>, <eot>, and <pad> during individual encoding, leading to the unintended repeated presence of their embeddings in the concatenated embedding. To address this, we conduct ablation experiments on whether to keep, remove, or replace special token embeddings. The final embedding excludes the <pad> embeddings and introduces a new unique <pad*> embedding to meet the target length. It retains all <sot> embeddings while removing all <eot> embeddings, resulting in the format "<sot> Text1. <sot> Text2. ... <pad*>". More details about this experiment can be found in Appendix A.

We can now use both CLIP and T5 for long-text encoding in T2I diffusion models. The next challenge is accurately representing all text segments in the generated images. To tackle this issue, we further fine-tune diffusion models with large-scale long texts paired with their corresponding images. We start with supervised training employing $\ell_2$ loss. However, we observe a clear optimization limit during training, with even the best version falling short of perfection (see the longSD(S) column in Table 2). This problem prompts us to explore additional preference optimization for long-text alignment alongside general supervised training, inspired by its success in LLMs (Ouyang et al., 2022).

### 3.2 SEGMENT PREFERENCE MODEL

In the context of preference optimization in T2I diffusion models, previous studies (Kirstain et al., 2023; Wu et al., 2023b) have employed CLIP-based human preference models to better align T2I

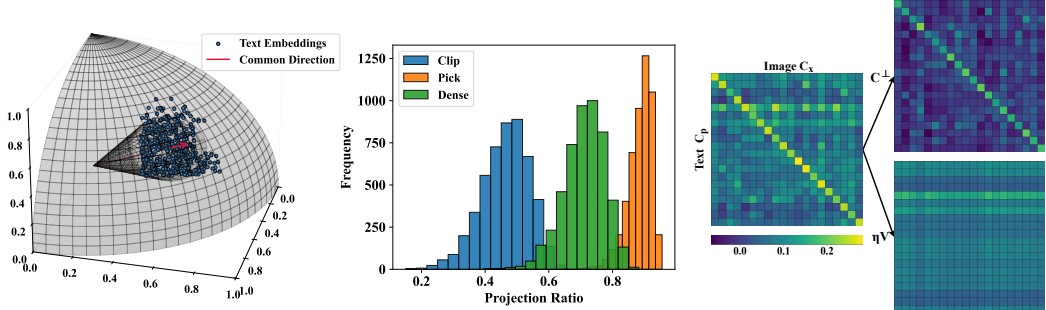

Figure 2: (a) Schematic results for text embeddings. (b) Statistics of the projection scalar $\eta$ for three models. (c) The relationship between the original score and the two scores after decomposition using our Denscore. In the three score tables, the diagonal represents the scores for paired data, while the off-diagonal positions indicate the scores for unpaired data.

diffusion models with human preferences. Since our objective—text alignment—is a key component of human preference, and CLIP-based structures are effective for large-scale training, we aim to adapt such preference models for our long-text alignment task.

The first step towards achieving the above goal is to enable CLIP-based preference models to accept long-text inputs. As shown in Section 2.2, these models introduce a new human preference training objective into the CLIP framework. However, they still have the same limited maximum input length and may struggle to reflect the varying impacts of different segments (e.g., sentences) with a single final score. To solve these problems, we split the long-text condition $p$ into $K$ segments, denoted as $\{\hat{p}_k\}_{k=1}^K$. Then, the new segment-level preference training loss is:

$$\mathcal{L}_{i \succ j}^{\text{seg}} = \frac{\exp(\sum_{k=1}^K \mathcal{R}(x_i, \hat{p}_k)/K)}{\exp(\sum_{k=1}^K \mathcal{R}(x_i, \hat{p}_k)/K) + \exp(\sum_{k=1}^K \mathcal{R}(x_j, \hat{p}_k)/K)}. \tag{6}$$

The above loss enables weakly supervised learning (Zhou, 2018) during training, eliminating the need for additional segment-level annotations. In comparison to its single-value counterpart, the new preference model supports long-text inputs and generates more detailed segment-level scores $\{\mathcal{R}(x, \hat{p}_k)\}_k$, along with an average final score $\sum_{k=1}^K \mathcal{R}(x, \hat{p}_k)/K = \mathcal{C}_X(x) \cdot \sum_{k=1}^K \mathcal{C}_P(\hat{p}_k)/K$. Thus, computing this average score is equivalent to first segment-level encoding the text input and using the average embedding, denoted as $C_P^{\text{seg}}(p) = \sum_{k=1}^K \mathcal{C}_P(\hat{p}_k)/K$, to compute the score. We refer to this score as Denscore, which will also function as a reward signal in the following section. If there is no confusion, we may omit the segment label in $C_P^{\text{seg}}(p)$.

## 4 PREFERENCE DECOMPOSITION

In this section, we explore preference optimization using the preference models mentioned above. We find that their preference scores comprise a text-relevant component and a text-irrelevant component, with the latter often causing overfitting in fine-tuning diffusion models. To address this, we propose a reweighting strategy for both components that reduces overfitting and enhances alignment.

### 4.1 ORTHOGONAL DECOMPOSITION

As shown in Section 2.2, the CLIP-based preference model $\mathcal{R}(x, p) = \mathcal{C}_X(x) \cdot \mathcal{C}_P(p)$ evaluates an image $x$ against a text $p$ with respect to human preferences. Some preferences concern whether the text condition $p$ is accurately represented in the image $x$, while others focus on visual factors such as photorealism and aesthetics, making them irrelevant to the text. We find a direct structural correspondence between these two types of preferences within the text embedding $\mathcal{C}_P(p)$. Specifically, different $\mathcal{C}_P(p)$ display a common direction, as illustrated in Figure 2 (a). This common direction corresponds to text-irrelevant preferences, while the remainder reflects text-relevant preferences.

To support this statement, we use text embeddings from a large prompt dataset $\mathcal{P}$ to compute the common text-irrelevant direction: $\mathbf{V} := \mathbb{E}_{p \sim \mathcal{P}} \mathcal{C}_P(p)/||\mathbb{E}_{p \sim \mathcal{P}} \mathcal{C}_P(p)||$. We then decompose the text embedding $\mathcal{C}_P(p)$ into two orthogonal parts: $\mathcal{C}_P^\perp(p) + \eta \mathbf{V}$, where $\eta = \mathcal{C}_P(p) \cdot \mathbf{V}$ is the projection scalar of $\mathcal{C}_P$ onto $\mathbf{V}$. For the value of $\eta$, we test CLIP, Pickscore and our Denscore, presenting the results in Figure 2 (b). We observe that $\mathcal{C}_P$ exhibits a strong positive scalar projection onto $\mathbf{V}$ ($\eta > 0.4$ for CLIP and $\eta > 0.6$ for the others), forming a core in representation space. The presence of $\mathbf{V}$ is referred to as the cone effect (Gao et al., 2019; Liang et al., 2022), which results from both model initialization and contrastive training.

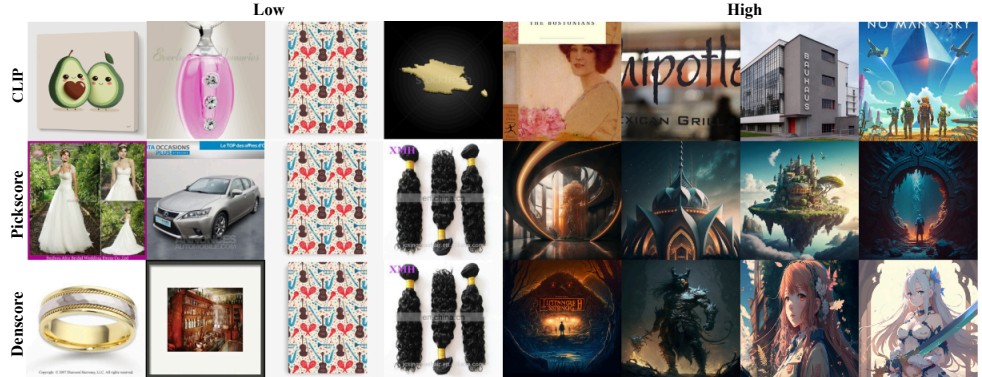

Figure 3: Retrieval results with low or high text-irrelevant scores, using three CLIP-based models.

To better understand the differing roles of these two components in the preference score $\mathcal{R}(x,t)$, we experiment with a 5k image-text dataset. Figure 2 (c) presents a score table illustrating the relationship between $\mathcal{C}_X(x) \cdot \mathcal{C}_P(p)$, $\mathcal{C}_X(x) \cdot \mathcal{C}_P^\perp(p)$ and $\mathcal{C}_X(x) \cdot \eta\mathbf{V}$. A more detailed version of this figure and real data statistics for the three scores can be found in Appendix B.2. For the second score $\mathcal{C}_X(x) \cdot \mathcal{C}_P^\perp(p)$, our experiments in Section 5.2 and Appendix B.3 show that it eliminates the influence of text-irrelevant components and focuses on measuring T2I alignment, which aligns with our objective. We refer to this aspect of Denscore as Denscore-O.

For the third score $\mathcal{C}_X(x) \cdot \eta\mathbf{V} = \eta(\mathcal{C}_X(x) \cdot \mathbf{V})$, the scalar $\eta$ is analyzed in the above two paragraphs. The remaining text-irrelevant term $\mathcal{C}_X(x) \cdot \mathbf{V}$ still needs further clarification. We provide images with low or high text-irrelevant scores in Figure 3 to visualize the score standard. Notably, the three models assign low scores to images that share similar characteristics, marked by dull visuals and large blank spaces. Conversely, while the high-scoring images selected by CLIP lack distinct features, the two preference models show strong alignment. Their high-scoring images feature rich details, well-organized layouts, and overall higher quality. This highlights the role of training on Equation 4, which provides a clear training objective—general human preference—for text-irrelevant scores. In contrast, CLIP's original contrastive training does not incorporate this aspect.

### 4.2 GRADIENT-REWEIGHT REWARD FINE-TUNING

To use the preference model $\mathcal{R}(x,p)$ as a reward signal for fine-tuning T2I diffusion models, there are two main challenges. One challenge is backpropagating the gradient through the entire multi-step iteration process. This can be addressed by the method mentioned in Section 2.3. However, previous works still struggle with overfitting, where fine-tuned images exhibit similar patterns (see Figure 6). While these patterns may enhance reward scores, they often lead to unsatisfactory images. We find that this is primarily because the text-irrelevant parts $\mathbf{V}$ constitute a significant portion of the optimization direction. Specifically, we fine-tune the generator $\epsilon_\theta$ using the reward signal $\mathcal{L}_r = 1 - \mathbb{E}_{x_T,p}\mathcal{R}(x_0^*, p)$, with the gradient calculated as follows:

$$\partial_\theta(1 - \mathbb{E}_{x_T,p}\mathcal{R}(x_0^*, p)) = -\partial_\theta\mathbb{E}_{x_T,p}(\mathcal{C}_P(p) \cdot \mathcal{C}_X(x_0^*)) = -\mathbb{E}_p(\mathcal{C}_P(p)^T\mathbb{E}_{x_T}\partial_\theta(\mathcal{C}_X(x_0^*)))$$
$$= -\mathbb{E}_p((\eta\mathbf{V} + \mathcal{C}_P^\perp(p))^T\mathbb{E}_{x_T}\partial_\theta(\mathcal{C}_X(x_0^*))), \tag{7}$$

where the item $\eta\mathbf{V} + \mathcal{C}_P^\perp(p)$ controls the gradient direction of $\epsilon_\theta$. According to Section 4.1, the text-irrelevant component $\mathbf{V}$ comprises a large portion of the entire item, overwhelming the gradient and producing similar output patterns regardless of the text input $p$. To mitigate this overfitting problem, we fine-tune the generator $\epsilon_\theta$ using a reweighted gradient:

$$\partial_\theta(1 - \mathbb{E}_{x_T,p}\mathcal{R}(x_0^*, p)) \approx -\mathbb{E}_p((\omega(\eta\mathbf{V}) + \mathcal{C}_P^\perp(p))^T\mathbb{E}_{x_T}\partial_\theta(\mathcal{C}_X(x_0^*))), \tag{8}$$

where $\omega$ is the reweighting factor for the common direction $\mathbf{V}$. This addresses the overfitting problem mentioned above. Additionally, this analysis clarifies why the original CLIP is ineffective for reward fine-tuning (see Figure 6). The issue arises because its text-irrelevant component $\mathbf{V}$ is not well-trained (see Section 4.1), resulting in an undefined optimization direction.

By combining the methods from the two sections above, we arrive at our complete *LongAlign* method. Detailed algorithms are available in Appendix A and C.

## 5 EXPERIMENT

In this section, we first provide our experimental setup, including models, training strategies, and evaluation metrics. We then present the results of our segment preference models. Next, we detail

Table 1: R@1 results for 5k text-to-image retrieval using different CLIP-based models.

| | CLIP-H | | HPSv2 | | Pickscore | | Denscore | |
|---|---|---|---|---|---|---|---|---|
| | Single | Average | Single | Average | Single | Average | Single | Average |
| $\mathcal{C}_P(p)$ | 86.10 | 80.40 | 42.34 | 16.72 | 54.00 | 31.84 | 83.96 | 75.90 |
| $\mathcal{C}_P^{\perp}(p)$ | 85.80 | 85.14 | 67.94 | 64.28 | 67.60 | 64.00 | 87.24 | **91.86** |

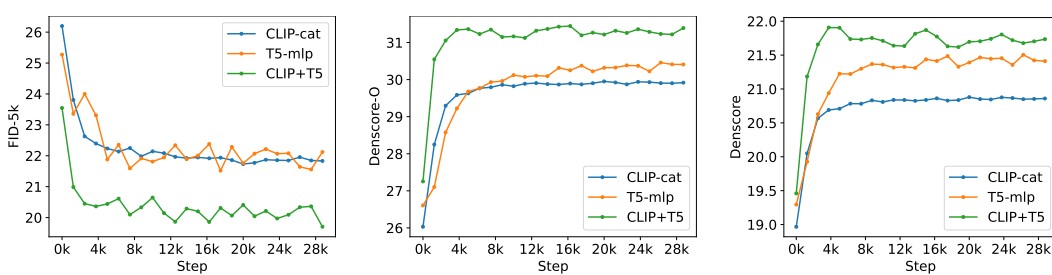

Figure 4: FID and Denscore results for diffusion models with different text encodings.

our long-text encoding results for diffusion models using various encoding methods. The advantages of gradient reweighting fine-tuning across different reward signals are then discussed. Finally, we present the generation results using our entire LongAlign alongside those of the baselines.

## 5.1 EXPERIMENTAL SETUP

**Model**. Our experiments cover three types of models: text encoders, Unets, and preference models. Specifically, we utilize the pretrained CLIP and T5 models as our text encoders. To align the embedding dimensions, we append a two-layer MLP to T5's output, following LaVi-Bridge (Zhao et al., 2024a). The Unet is derived from the pretrained Stable Diffusion v1.5[1]. Instead of full fine-tuning, we fine-tune the Unets using LoRA with a rank of 32 and update both the ResNet blocks and attention blocks within the Unets. For the preference model, we select two pretrained models: Pickscore and HPSv2. Additionally, we introduce our new segment-level preference model, Denscore. All three models are fine-tuned from pretrained CLIP models.

**Training**. (1) For training the Unet, we utilize a dataset of approximately 2 million images, including 500k from SAM (Kirillov et al., 2023), 100k from COCO2017 (Lin et al., 2014), 500k from LLaVA (a subset of the LAION/CC/SBU dataset), and 1 million from JourneyDB (Sun et al., 2024). We randomly reserve 5k images for evaluation. All images are recaptioned using LLaVA-Next (Liu et al., 2023) or ShareCaptioner (Chen et al., 2023b) and resized to $512 \times 512$ pixels. We optimize the model using the AdamW optimizer with a learning rate of $3 \times 10^{-5}$, a 2k-step warmup, and a total batch size of 192. Training is conducted on 6 A100-40G GPUs for 30k steps over 12 hours. (2) For the reward fine-tuning (RFT) stage of the Unet, we use the same settings as before but with a batch size of 96 and 4k total training steps over 8 hours. (3) For training the segment preference model, we use the same settings as for Pickscore (Kirstain et al., 2023), employing CLIP-H on Pickscore's training data, along with LLaVA-Next captions and our new segment-level loss function. More details about training the preference model can be found in Appendix B.1.

**Evaluation**. We evaluate our methods using the FID, Denscore, Denscore-O and VQAscore (Lin et al., 2024) metrics on the 5k-image evaluation dataset. FID evaluates the distribution distance between the dataset and generated images. Denscore assesses human preference for generated images, while Denscore-O and VQAscore focuses on the text alignment of those images. Additionally, we employ GPT-4o (OpenAI, 2024) to evaluate 1k images against baselines, mitigating the risk of overfitting to Denscore. The GPT-4o evaluation template can be found in Appendix D. All our experiments employ UniPC (Zhao et al., 2024b) with 25-step sampling, maintaining a consistent classifier-free guidance factor as per the original papers. In Appendix C.3, we also evaluate our models on two additional benchmarks, including DPG-Bench (Hu et al., 2024), to assess our T2I generation performance on prompts of varying structures and lengths.

## 5.2 SEGMENT PREFERENCE MODEL

To demonstrate that our analysis of the preference model is generally applicable, we compare four CLIP-based models: the pretrained CLIP, the single-value preference models Pickscore and HPSv2,

---

[1]https://huggingface.co/stable-diffusion-v1-5/stable-diffusion-v1-5

Table 2: FID and Denscore results for $512 \times 512$ image generation using different models. PlayG-2, KanD-2.2 and ELLA are from Li et al. (2024a), Razzhigaev et al. (2023) and Hu et al. (2024).

| Model | SD-1.5 | SD-2.1 | PlayG-2 | PixArt-$\alpha$ | KanD-2.2 | ELLA | longSD (S) | longSD (S+R) |
|---|---|---|---|---|---|---|---|---|
| FID-5k | 24.96 | 25.80 | 23.92 | 22.36 | 20.04 | 24.38 | 20.09 | **19.63**/24.28 |
| Denscore-O | 29.20 | 30.15 | 28.80 | 33.48 | 33.30 | 32.92 | 31.29 | 32.83/**35.26** |
| Denscore | 20.29 | 20.91 | 21.22 | 22.78 | 22.70 | 22.11 | 21.72 | 22.74/**23.79** |
| VQAscore | 84.57 | 85.61 | 85.26 | 86.96 | 86.31 | 86.85 | 86.18 | 87.11/**87.24** |

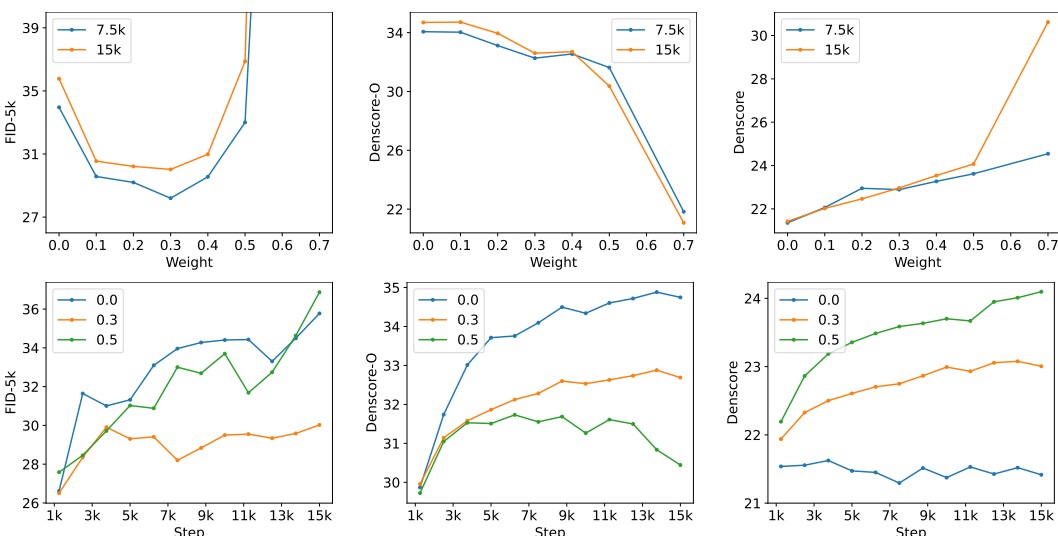

Figure 5: FID and Denscore results for diffusion models using different gradient reweighting factors.

as well as our new segment-level preference model, Denscore. According to Section 4.1, the text-irrelevant embedding $\mathbf{V}$ reflects the text-irrelevant preference, while the text-relevant embedding $\mathcal{C}_P^\perp(p)$ reflects the T2I alignment, which is our objective.

To assess the ability of the text-relevant part $\mathcal{C}_P^\perp(p)$, we provide the R@1 retrieval accuracy for the four models on the 5k evaluation dataset in Table 1. Here, we either utilize the embedding encoding from the first 77 tokens in a single-pass approach (Single) or take the average of segment-level embeddings $\mathcal{C}_P^{\text{seg}}(p)$ (Average). The final score is computed using the image embedding $\mathcal{C}_X(x)$ with either the full text embedding $\mathcal{C}_P$ or solely the text-relevant part $\mathcal{C}_P^\perp(p)$. We find that: (1) Across all preference models, the text-relevant embedding outperforms the original embedding, as it eliminates the impact of text-irrelevant factors. (2) Only Denscore with segment-level training yields better segment-level retrieval accuracy, while the other models don't benefit from, and are confused by, the extra information provided by averaging multiple embeddings. (3) Although T2I alignment is only a partial optimization objective in Denscore, the Denscore model outperforms CLIP. This highlights the importance of segment-level encoding strategies and preference decomposition.

In Appendix B.3, to better evaluate Denscore's performance under varying text lengths, we employ different maximum sentence prompts to obtain R@1 retrieval accuracy. Additionally, we conduct an experiment in the same appendix to identify specific misaligned segments in long inputs, showing that segment-level scoring provides more detailed information.

## 5.3 LONG-TEXT ENCODING

Here, we provide FID and Denscore results for supervised fine-tuning (SFT), comparing various text encoding methods: CLIP with concatenation (CLIP-cat), T5 with an additional two-layer MLP (T5-mlp)[2], and a combination of CLIP and T5 (CLIP+T5). The results in Figure 4 show that CLIP-cat and T5-mlp perform similarly, while the CLIP+T5 model significantly outperforms them. This suggests that the CLIP+T5 model is preferable, as it leverages the strengths of both CLIP's image-text paired pretraining and T5's pure long-text encoding capabilities. This finding is consistent with

---

[2]We utilize LaVi-bridge's (Zhao et al., 2024a) pretrained MLP, which means we only include the short-to-long fine-tuning.

Origin                 Reweight (ours)

CLIP    HPSv2    Pickscore    Denscore      CLIP    HPSv2    Pickscore    Denscore

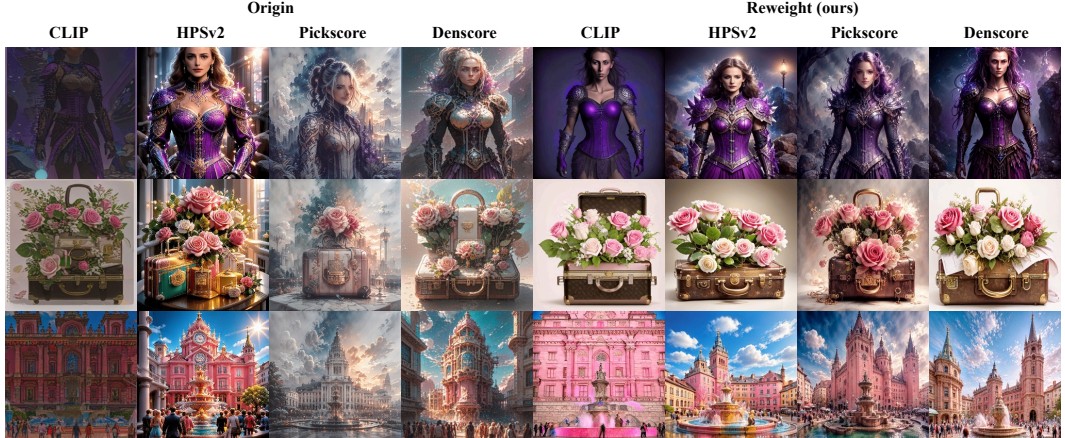

Figure 6: Generation results using different reward signals, with and without gradient reweighting. The corresponding text conditions can be found in Appendix F.

the results of previous works (Saharia et al., 2022; Li et al., 2024b), while these earlier models still face input length limitations due to CLIP's maximum token count.

## 5.4 REWARD FINE-TUNING

To simplify and accelerate ablation studies of reward fine-tuning (RFT), we leverage both LCM-LoRA (Song et al., 2023; Luo et al., 2023) and DRTune (Wu et al., 2024) to speed up fine-tuning using only CLIP-cat encoding for optimal strategy identification in this subsection.

**Weighting Factor**. In Figure 5, we provide the FID and Denscore results for different gradient reweight factors $\omega$ and training steps. According to the first three experiments with different reweight factors, the FID results exhibit a parabolic shape, with $\omega = 0.3$ achieving the best FID results. In the last three experiments with different training steps, only $\omega = 0.3$ maintains relatively stable FID results while simultaneously improving both Denscore and Denscore-O results. In contrast, the other two options improve only one of the Denscore or Denscore-O metrics, accompanied by a significant increase in FID, indicating an apparent overfitting problem. Additionally, please note that the optimal value of $\omega$ can vary depending on the model and training strategy used.

**Reward Signal**. In Figure 6, we provide visual results showing the benefits of gradient reweighting on reward signals from various preference models, including CLIP, HPSv2, Pickscore, and our Denscore. Notably, this method also partially addresses the limitation of using reward signals from pretrained CLIP. However, CLIP cannot yet outperform preference models because its text-irrelevant component **V** is not well-trained, which aligns with our analysis in Section 4.2.

## 5.5 GENERATION RESULT

**Foundation Model**. In this subsection, we apply our entire LongAlign approach to train our long Stable Diffusion (longSD) and compare it against other baselines. Here, we present the results of SFT (S) at 28k steps and SFT+RFT (S+R) with $\omega = 0.3$, at 1.25k and 3.75k steps. As shown in Table 2, SFT+RFT clearly outperforms both the original SD-1.5 and the basic SFT version. Compared to other advanced foundation models, our longSD model surpasses them in terms of FID and Denscore metrics. Furthermore, we use GPT-4o (OpenAI, 2024) to evaluate 1k longSD(S+R) results, comparing them to baselines and mitigating the risk of overfitting. The new results (see

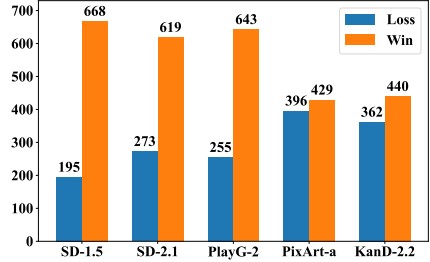

Figure 7: GPT-4o evaluation results of T2I alignment across different models.

Figure 7) are consistent with our previous scores. We also provide visualizations in Figure 8. All these findings highlight the effectiveness of our method for generating high-quality images from long texts, demonstrating significant potential beyond altering the model structure.

**Alignment Strategy**. In addition to varying model structures, there are other methods that improve T2I alignment. Some of these approaches incorporate the assistance of LLMs, such as Ranni (Feng et al., 2024) and RPG-Diffusers (Yang et al., 2024), while others employ improved training strategies, such as Paragraph-to-Image (P2I) diffusion (Wu et al., 2023a). LLM-based methods that

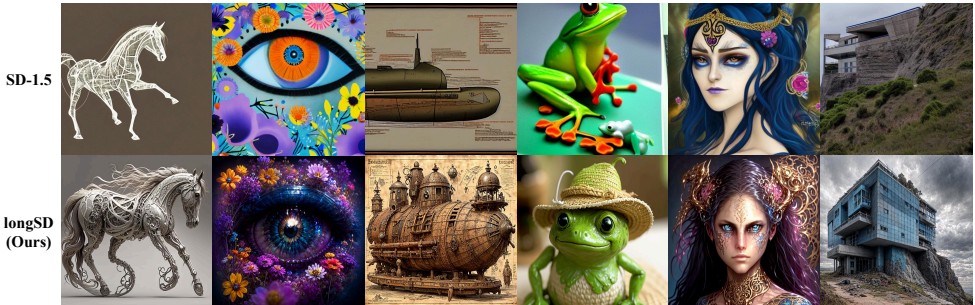

Figure 8: The generation results show a comparison of performance before and after our fine-tuning on SD-1.5. The corresponding text conditions are provided in Appendix F.

involve additional LLM assistance increase computational requirements and encounter significant out-of-distribution (OOD) issues with long-text inputs. More details on OOD issues can be found in Appendix E. Regarding improved training strategies, our approach is orthogonal to P2I diffusion, so we compare the original P2I diffusion with its fine-tuned version using our method. The results in Appendix D Table 11 also show clear improvements in terms of FID, Denscore, and GPT-4o.

## 6    RELATED WORK

**Diffusion-based Generation**. The popularity of diffusion models has surged, driven by breakthroughs in fast sampling methods (Song et al., 2021a; Lu et al., 2022; Zhang & Chen, 2022) and text-conditioned generation (Saharia et al., 2022; Chen et al., 2023a). Moreover, cascaded and latent space models (Ho et al., 2022; Rombach et al., 2022) have enabled the creation of high-resolution images. These advancements have also unlocked various applications beyond T2I generation, including the ability to create content with a consistent style (Hertz et al., 2023) and image editing tools (Hertz et al., 2022; Brooks et al., 2023). Furthermore, advancements in foundation models like U-ViT (Bao et al., 2023) and DiT (Peebles & Xie, 2023) suggest even greater potential.

**Text-to-Image Evaluation**. Traditional metrics, such as Inception Score (IS) (Salimans et al., 2016) and Fréchet Inception Distance (FID) (Heusel et al., 2017), have limitations in evaluating the quality of T2I generation. To address these limitations, three approaches have emerged. One approach employs Perceptual Similarity Metrics like LPIPS (Zhang et al., 2018), which utilize pretrained models to assess image similarity from a human perspective. Another approach involves using detection models (Huang et al., 2023) to extract and analyze key object alignments. A third approach fine-tunes human preference models (Xu et al., 2024; Kirstain et al., 2023; Wu et al., 2023b) on datasets of human preferences for images generated from specific prompts, effectively turning them into proxies for human evaluation.

**Reward Fine-tuning**. Training text-to-image models with a reward signal can be effective in targeting specific outcomes. Two primary approaches have emerged. One leverages reinforcement learning to optimize rewards that are difficult to calculate using traditional methods, such as DPOK (Fan et al., 2024) and DDPO (Black et al., 2023). Another approach, exemplified by DiffusionCLIP (Kim et al., 2022), DRaFT (Clark et al., 2023) and AlignProp (Prabhudesai et al., 2023), involves back-propagation through sampling. This method leverages human preferences and other differentiable reward signals to optimize diffusion models for specific targets. Recently, DRTune (Wu et al., 2024) further improves training speed by stopping the gradient of the diffusion model's input.

## 7    DISCUSSION

In this paper, we propose LongAlign to enhance long-text to image generation. We examine segment-level text encoding strategies for processing long-text inputs in both T2I diffusion models and preference models. In addition, we enhance the role of preference models by analyzing their structure and decomposing them into text-relevant and text-irrelevant components. During reward fine-tuning, we propose a gradient reweighting strategy to reduce overfitting and enhance alignment. In our experiments, we utilize the classical SD-1.5 and effectively fine-tune it to outperform stronger foundation models, demonstrating significant potential beyond designing new model structures.

The limitation of this paper is that, while we have made improvements, we still do not fully address the generation of long prompts with complex contextual dependencies or those requiring strong semantic understanding, partly due to CLIP's constraints. In the future, we will explore more powerful training strategies beyond using CLIP-based preference models.

ACKNOWLEDGMENTS

This work is partially supported by the Hong Kong Research Grants Council (RGC) General Research Fund (17203023), the RGC Collaborative Research Fund (C5052-23G), the National Natural Science Foundation of China/RGC Collaborative Research Scheme (CRS_HKU703/24), the Hong Kong Jockey Club Charities Trust under Grant 2022-0174, UBTECH Robotics funding, and The University of Hong Kong's Startup and Seed Funds for Basic Research for New Staff. Additionally, C. Li is supported by the National Natural Science Foundation of China (No. 92470118), Beijing Natural Science Foundation (No. L247030), and Beijing Nova Program (No. 20230484416).

ETHICS STATEMENT

This research acknowledges the ethical implications of advancements in text-to-image (T2I) diffusion models. We commit to responsible practices by ensuring transparency in our methodologies and findings. Our segment-level encoding and preference optimization techniques are designed to enhance alignment while minimizing bias and overfitting.

We prioritize inclusivity and societal values by actively engaging with diverse stakeholders to understand the cultural impacts of our work. Our ongoing commitment to ethical research practices aims to ensure that our technologies contribute positively to society while mitigating potential harms.

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

## A CONCATENATION STRATEGY

For the CLIP encoder in T2I diffusion models, we first show the pipeline of our new segment-level encoding in Figure 9. In this pipeline, segment-level encoding and concatenation may seem straightforward, but the optimal concatenation strategy remains unclear, as shown in Figure 10. This is because each segment contains special tokens, such as <sot>, <eot>, and <pad>, leading to their corresponding embeddings' unintended repeated presence in the final concatenated embedding. This raises the question of whether to retain, remove, or replace them in the final embeddings.

For <pad> tokens in each segment, we omit them in the final embeddings since they lack alignment information. However, we sometimes need to introduce a new <pad> embedding to ensure aligned token sequence lengths. To address this, we assign a unique embedding to each required <pad> token's position. Specifically, our unique <pad> embedding is the average value of all <pad> tokens of an empty sentence, which we denote as <pad*>. For <sot> and <eot> tokens, we then experiment with pretrained diffusion models to find the optimal strategy. As shown in Figure 10, our results indicate that the optimal approach is to keep all <sot> token embeddings and remove all <eot> token embeddings. Therefore, our final embeddings used in this paper take the form "<sot> Text1. <sot> Text2. ... <pad*>".

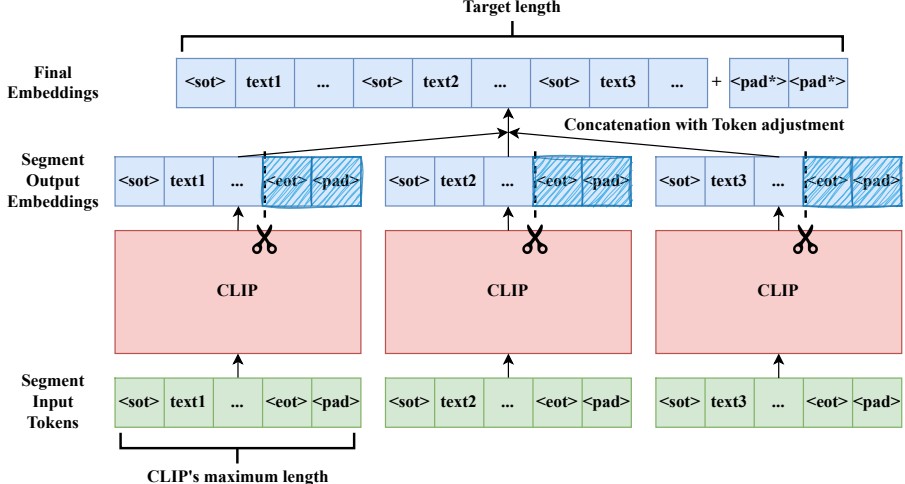

Figure 9: The visualization of our new segment-level text encoding for diffusion models is presented.

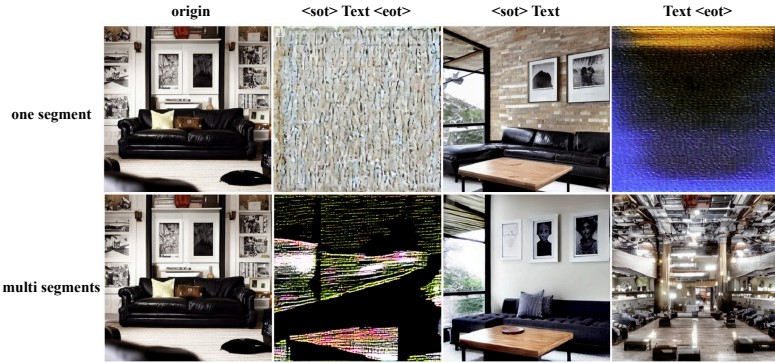

Figure 10: Generation results under different embedding concatenation strategies.

The last column of the experiment above shows that removing all <sot> tokens destroys the generated results. In Figure 11, we also compare the effects of keeping only the first <sot> token (1-sot) versus retaining all <sot> tokens (ours). Additionally, we investigate different segmentation strategies by testing the difference between treating each sentence as a segment (ours) and grouping several consecutive sentences into a segment (multi), provided their total token count remains under 77. We find that these two new ablation studies do not show significant differences in the results.

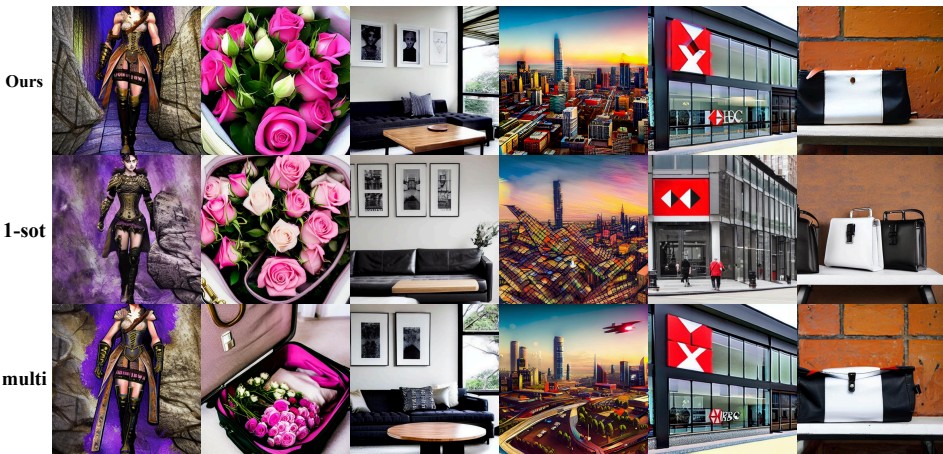

Figure 11: Generation results of SD-1.5 using various segment encoding strategies.

Furthermore, in Figure 12, we analyze the cross-attention map for both original and segment-level encodings. The results show similar interaction behaviors between them. When the input prompt labels objects (e.g., dog and duck, clock and pen) and references them across different segments, the model accurately aligns these objects across those segments. This demonstrates that T2I diffusion models with segment encoding can handle cross-sentence information.

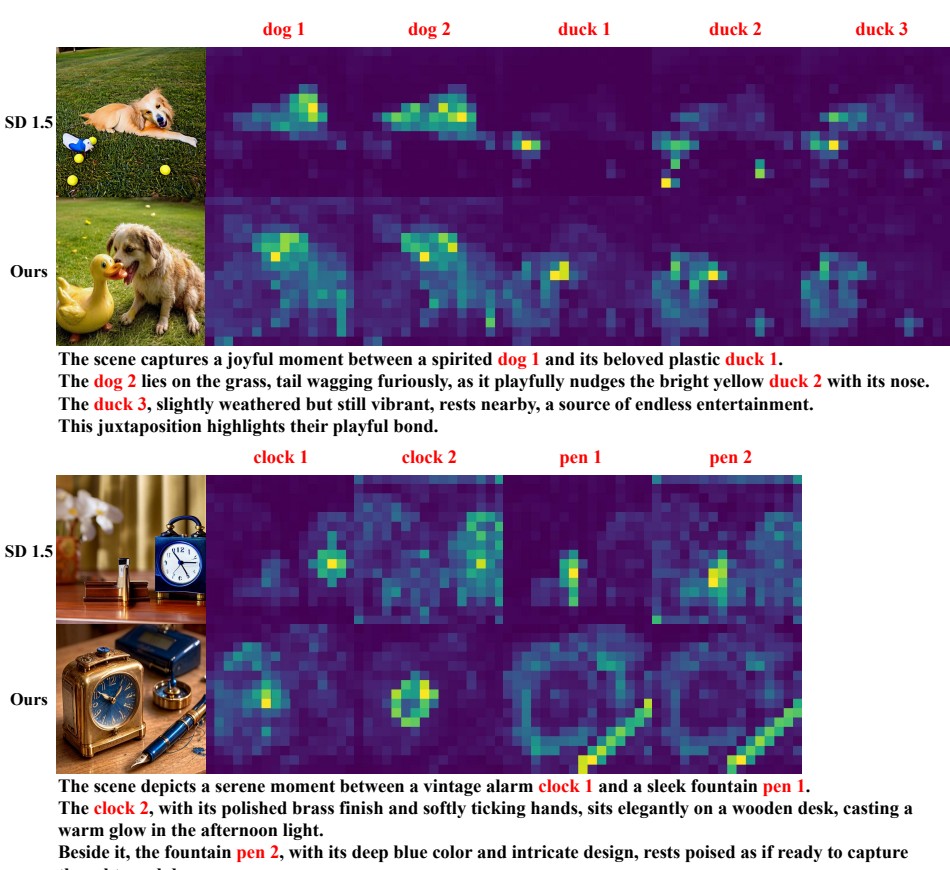

Figure 12: The visualization of the generation results and cross-attention maps for both original and segment-level encodings using SD-1.5 and our fine-tuned version. The 1,2,3 behind the nouns are for readability and do not be included in the input prompt.

# B SEGMENT PREFERENCE MODEL

Here, we provide a detailed analysis of our segment preference models with respect to training, visualization, and evaluation.

## B.1 TRAINING

We train a segment preference model by incorporating long and detailed text conditions generated by LLaVA-Next alongside our segment-level preference loss function. Based on the analysis in Section 4.1, we choose to combine the segment-level loss for refining text-relevant aspects with the original loss for improving aspects unrelated to text, such as aesthetics. The loss function $\mathcal{L}_{i \succ j}^{\text{seg-a}}$ (where "a" denotes addition) is:

$$\mathcal{L}_{i \succ j}^{\text{seg-a}} = \mathbb{E}_{x,\{\hat{p}_k\}} \sigma \left( \sum_{k=1}^{K} (\mathcal{C}_X(x_i) \cdot \mathcal{C}_P(\hat{p}_k)/K) - \sum_{k=1}^{K} (\mathcal{C}_X(x_j) \cdot \mathcal{C}_P(\hat{p}_k)/K) \right) + \\ \mathbb{E}_{x,p} \sigma(\mathcal{C}_X(x_i) \cdot \mathcal{C}_P(p) - \mathcal{C}_X(x_j) \cdot \mathcal{C}_P(p)),$$

(9)

where $\sigma(x) = \frac{1}{1+e^{-x}}$ is the sigmoid function and $\frac{e^m}{e^m+e^n} = \sigma(m-n)$. In addition, we substitute $\mathcal{C}_P(\hat{p}_k)$ with $\mathcal{C}_P^{\perp}(\hat{p}_k)$ to help the new segment-level loss focus on the T2I alignment part and avoid influencing the text-irrelevant part. The new loss function $\mathcal{L}_{i \succ j}^{\text{seg-o}}$ (where "o" means orthogonal) is:

$$\mathcal{L}_{i \succ j}^{\text{seg-o}} = \mathbb{E}_{x,\{\hat{p}_k\}} \sigma \left( \sum_{k=1}^{K} (\mathcal{C}_X(x_i) \cdot \mathcal{C}_P^{\perp}(\hat{p}_k)/K) - \sum_{k=1}^{K} (\mathcal{C}_X(x_j) \cdot \mathcal{C}_P^{\perp}(\hat{p}_k)/K) \right) + \\ \mathbb{E}_{x,p} \sigma(\mathcal{C}_X(x_i) \cdot \mathcal{C}_P(p) - \mathcal{C}_X(x_j) \cdot \mathcal{C}_P(p)),$$

(10)

where $\{\hat{p}_k\}$ represents the segments split from the long text generated by LLava-Next, and $p$ is the original short text.

If we prioritize alignment, $\mathcal{L}_{i \succ j}^{\text{seg-a}}$ is the better choice; if we want to balance both alignment and aesthetics, $\mathcal{L}_{i \succ j}^{\text{seg-o}}$ is the preferable option. This is because Equation 10 eliminates the influence of the first loss item on the text-irrelevant part $\mathbf{V}$, allowing it to better focus on factors unrelated to the text, such as aesthetics. To support this, we present retrieval results with the highest scores used text-irrelevant components $\mathbf{V}$ trained with two different loss functions. The results from Equation 10 align more closely with human preferences. Therefore, in this paper, we choose $\mathcal{L}_{i \succ j}^{\text{seg-o}}$ in Equation 10 to train our new segment preference model.

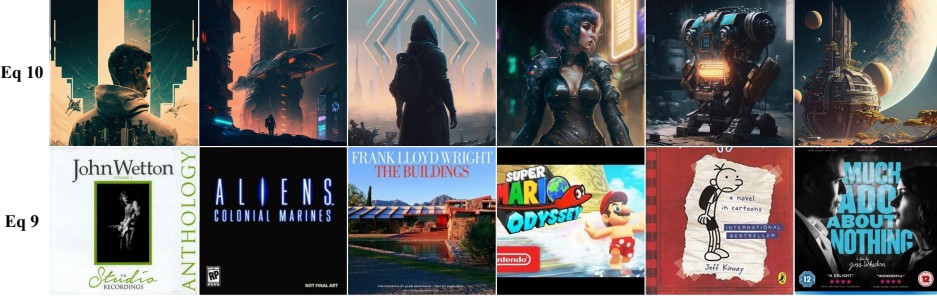

Figure 13: The retrieval results with the highest scores used text-irrelevant components $\mathbf{V}$ trained with two different loss functions.

## B.2 VISUALIZATION OF SCORE

After training the new segment preference models, we want to analyze the behavior of three scores $\mathcal{C}_X(x) \cdot \mathcal{C}_P(p)$, $\mathcal{C}_X(x) \cdot \mathcal{C}_P^{\perp}(p)$ and $\mathcal{C}_X(x) \cdot \eta \mathbf{V}$. We are interested in the relationships among them, and additionally, in the differences in these scores between different CLIP-based models. Here, we choose CLIP, Pickscore and our Denscore as our experimental targets, as they all use the same model structure, while the last two are fine-tuned on Pickscore's human preference dataset. We present the

results in Figures 14 and 15. The first figure uses 20 data pairs to visualize the relationships and differences, while the second figure displays the actual statistics based on 5,000 data pairs.

For the original score $\mathcal{C}_X(x) \cdot \mathcal{C}_P(p)$, only the original CLIP behaves in such a way that its score can be close to zero when the image and text inputs are unpaired, while the other two models still assign relatively high scores to such unpaired inputs. This is because the preference models assess the image not only based on text alignment but also on other purely visual factors, such as aesthetics.

For the text-relative score $\mathcal{C}_X(x) \cdot \mathcal{C}_P^\perp(p)$, after removing the influence of the text-irrelevant part, all three models provide nearly zero scores for unpaired input data, which supports our analysis that this score focuses on the T2I alignment and explains why this score achieves the best retrieval results for preference models, as shown in Table 1.

For the third score $\mathcal{C}_X(x) \cdot \eta \mathbf{V}$, we have analyzed the scalar $\eta$ in the main paper, which is determined entirely by the text input and is image-irrelevant. Here, we focus on the text-irrelevant score $\mathcal{C}_X(x) \cdot \mathbf{V}$. We find that this score is strongly positive for the two preference models. This corresponds to pure visual factors in preference, as shown in Figure 3. On the other hand, for the original CLIP, this score is nearly zero. This indicates that the common direction of text embedding is almost orthogonal to the image embeddings, which do not contribute to the final score of CLIP.

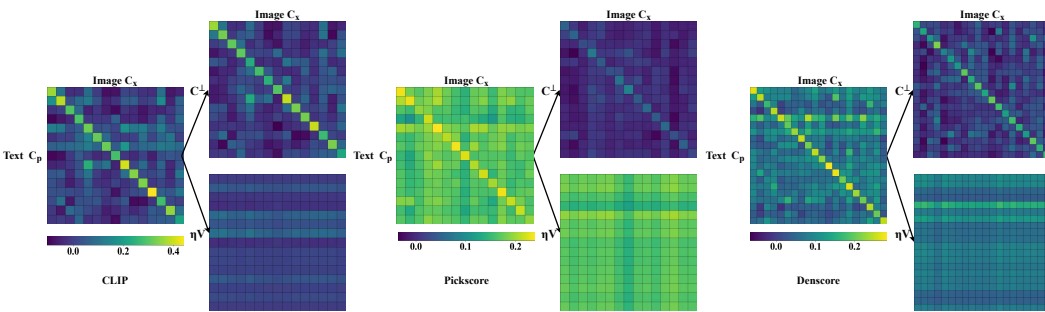

Figure 14: Logit results for different models, both before and after orthogonal decomposition.

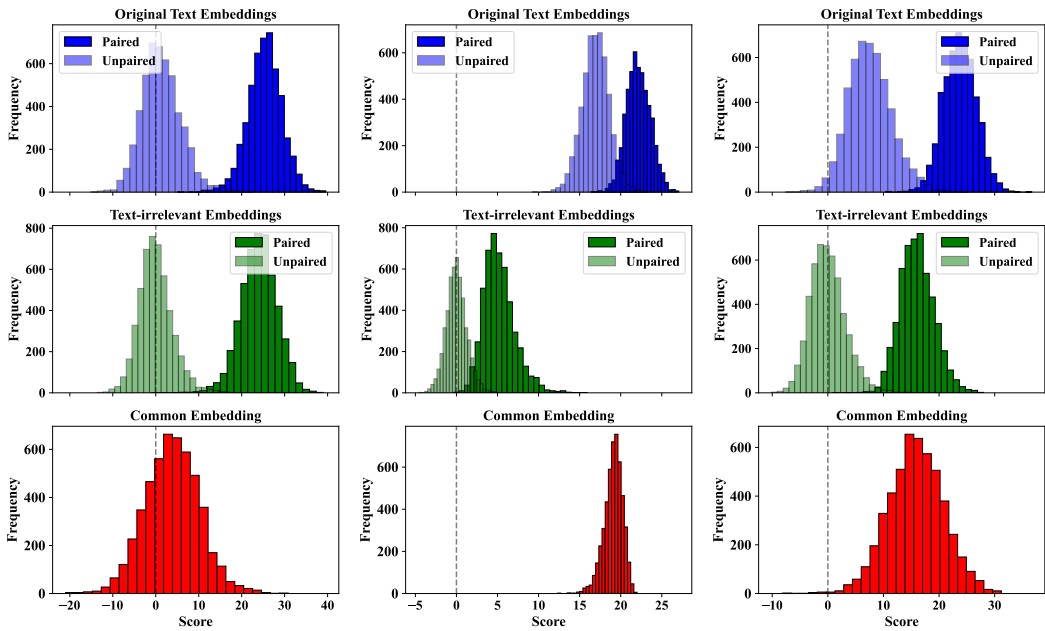

Figure 15: The real data statistics for the diagonal paired data and the off-diagonal unpaired data.

### B.3 Evaluation

To better support the above analysis, we design two additional experiments to test the effectiveness of our score alongside the experiment in Section 5.2.

Firstly, to evaluate Denscore's performance under varying text lengths, we employ different maximum sentence prompts to obtain R@1 retrieval accuracy, as shown in Table 3. We find that our Denscore, using segment-level training, consistently outperforms the others, except in the one-sentence setting. In this one-sentence setting, our segment-level training becomes less meaningful, but our results still outperform other existing preference models.

Additionally, to identify specific misaligned segments in long inputs, we conduct an experiment, illustrated in Figure 16, to show that segment-level scoring provides more detailed information. In some cases, certain segments align better with the first image, while other segments align better with the second image. This makes the overall score relatively meaningless, while segment-level scores continue to perform well.

Table 3: R@1 results for 5k text-to-image retrieval with varying maximum numbers of sentences.

| max number | 1 | 2 | 3 | 4 | 6 | 8 |
|---|---|---|---|---|---|---|
| CLIP | **53.06** | 70.90 | 76.70 | 79.62 | 83.00 | 84.12 |
| HPSv2 | 41.86 | 53.66 | 56.48 | 59.14 | 62.58 | 63.96 |
| Pickscore | 42.34 | 53.86 | 57.56 | 60.22 | 63.60 | 63.54 |
| Denscore | 52.72 | **72.70** | **78.78** | **83.10** | **88.16** | **89.94** |

## C Decomposed preference optimization

### C.1 Pseudocode

Here, we provide the pseudocode in Algorithm 1 for the entire decomposed preference optimization pipeline discussed in this paper.

---

**Algorithm 1** Decomposed Preference Optimization for T2I Diffusion Models

---

1: **Input:** Long-text input $T$, Initial T2I diffusion model $M$, Preference model $R$
2: **Output:** Fine-tuned T2I diffusion model $\hat{M}$
3: $S \leftarrow \text{Segment}(T)$ {Step 1: Divide $T$ into segments (e.g., sentences) as $S = \{s_1, s_2, \ldots, s_n\}$}
4: **for** each segment $s_i \in S$ **do**
5: $\quad E_i \leftarrow \text{Encode}(s_i)$ {Step 2: Encode segment $s_i$ (as shown in Section 3.1 and Figure 9)}
6: **end for**
7: $E \leftarrow \text{Concatenate}(E_1, E_2, \ldots, E_n)$ {Step 3: Concatenate segment embeddings (as shown in Section 3.1 and Figure 9)}
8: $I \leftarrow M(E)$ {Step 4: Generate image $I$ from embeddings $E$ using the T2I diffusion model}
9: $E_{\text{image}}, E_{\text{segment}} \leftarrow R(I, S)$ {Step 5: Compute the Preference Embeddings of segments $S$ and image $I$ (as shown in Section 3.2)}
10: $E_{\text{overall}} \leftarrow \frac{1}{n} \sum_{i=1}^{n} E_{\text{segment}}$ {Compute overall average Embedding}
11: $(E_{\text{text-relevant}}, E_{\text{text-irrelevant}}, \eta_{\text{scaler}}) \leftarrow \text{Decompose}(E_{\text{overall}})$ {Step 6: Decompose overall score into relevant and irrelevant components (as shown in Section 4.1)}
12: $L_{\text{loss}} \leftarrow E_{\text{image}} \cdot E_{\text{text-relevant}} + \omega_{\text{irrelevant}} E_{\text{image}} \cdot \eta_{\text{scaler}} E_{\text{text-irrelevant}}$ {Step 7: Calculate adjusted loss (as shown in Section 4.2)}
13: $\hat{M} \leftarrow \text{FineTune}(M, L_{\text{loss}})$ {Step 8: Fine-tune T2I model using computed loss (as shown in Section 2.3)}
14: **Return** $\hat{M}$

---

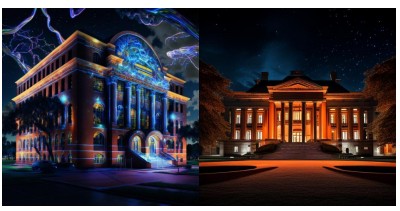

The image depicts a large, ornate building at night, illuminated by a combination of **natural and artificial light**. [29.296875, 25.976562] The building has a classical architectural style, with columns and arches, and is **bathed in a warm, orange glow** that highlights its facade. [20.214844, 26.757812] In front of the building, there is **a well-maintained lawn with a path leading up to the entrance**. [10.498047, 15.820312] The overall atmosphere of the image is one of grandeur and mystery, enhanced by **the dramatic lighting and the fantastical elements in the sky**. [22.949219, 17.578125]

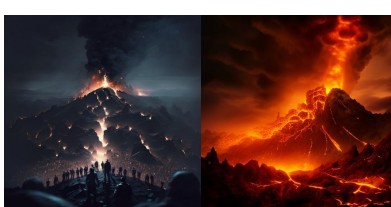

The image depicts **a dramatic and apocalyptic scene.** [26.367188, 20.800781] At the center of the image, there is **a massive volcanic eruption with a bright orange and yellow lava flow** cascading down the sides of a mountain. [20.507812, 29.101562] The sky is filled with dark clouds and smoke, suggesting a catastrophic event. [22.65625, 20.800781] In the foreground, there is **a group of people standing on a rocky outcropping**. [20.117188, 8.398438]

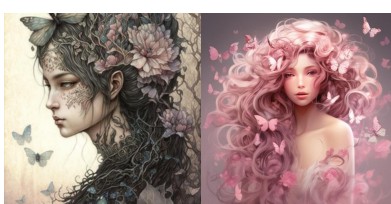

The image is a digital artwork that features a stylized female figure with a fantastical and ethereal appearance. [28.101562, 26.5625] The figure has long, flowing hair that cascades down her back, intertwined **with what appears to be delicate, pink flowers**. [33.789062, 31.835938] The figure's face is characterized by a pale complexion, large, expressive eyes, and a subtle, serene expression. [20.703125, 20.507812] Her lips are parted slightly, and **her gaze is directed off to the side**, giving the impression of contemplation or daydreaming. [21.679688, 18.945312]

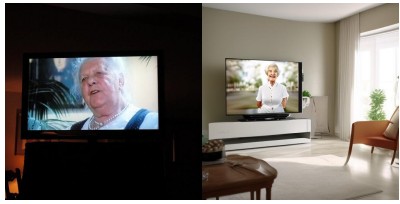

The image is dominated by a television screen mounted on the wall. [25.78125, 23.339844] The screen is alive with the image of an elderly woman, who is elegantly dressed in a white blouse and a pearl necklace. [23.632812, 24.316406] The room itself is **dimly lit, creating an atmosphere of tranquility**. [20.800781, 14.257812] The television, being the central object, draws the viewer's attention, while **the plant in the background adds depth to the scene**. [24.21875, 28.125]

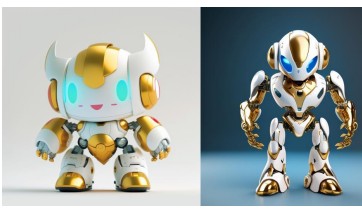

The image features a 3D rendering of a robot character. [29.101562, 28.710938] The robot has a predominantly white body with gold accents, including a gold chest plate and gold accents on its arms and legs. [33.007812, 33.203125] The robot's eyes are large and round, with **a blue light inside them**. [20.800781, 22.558594] The background of the image is **a plain, light gray color**, providing a neutral backdrop that contrasts with the robot's colorful design. [25.78125, 20.214844]

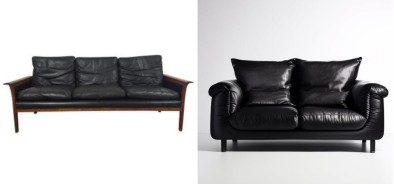

The image presents a scene dominated by a black leather sofa, which is the central object in the frame. [27.148438, 26.953125] The sofa, **with its three cushions**, is positioned against a stark white background, creating a striking contrast. [29.882812, 28.320312] The cushions, like the sofa, are black and appear to be made of leather, suggesting a uniform color scheme throughout the piece. [26.5625, 27.929688] The sofa is **designed with a wooden frame**, adding a touch of warmth to the otherwise monochrome setting. [24.511719, 20.117188]

Figure 16: Identifying the best-aligned image and segment pairs using segment-level preference scores.

## C.2 OPTIMIZATION VISUALIZATION

In Figure 17, we present additional visualizations of the generation results with and without our reweighting strategies at various ratios to demonstrate the effectiveness of our method. A ratio of 1 indicates the original loss, resulting in significant overfitting, where all images exhibit similar patterns regardless of the inputs. A ratio of 0 means that the loss only considers the text-relevant part, leading to low image quality that does not align with human preferences. We observe that a ratio of 0.3 yields the best image quality. These experiments and the results in Section 5.4 demonstrate that our reweighting strategy effectively reduces overfitting and improves alignment.

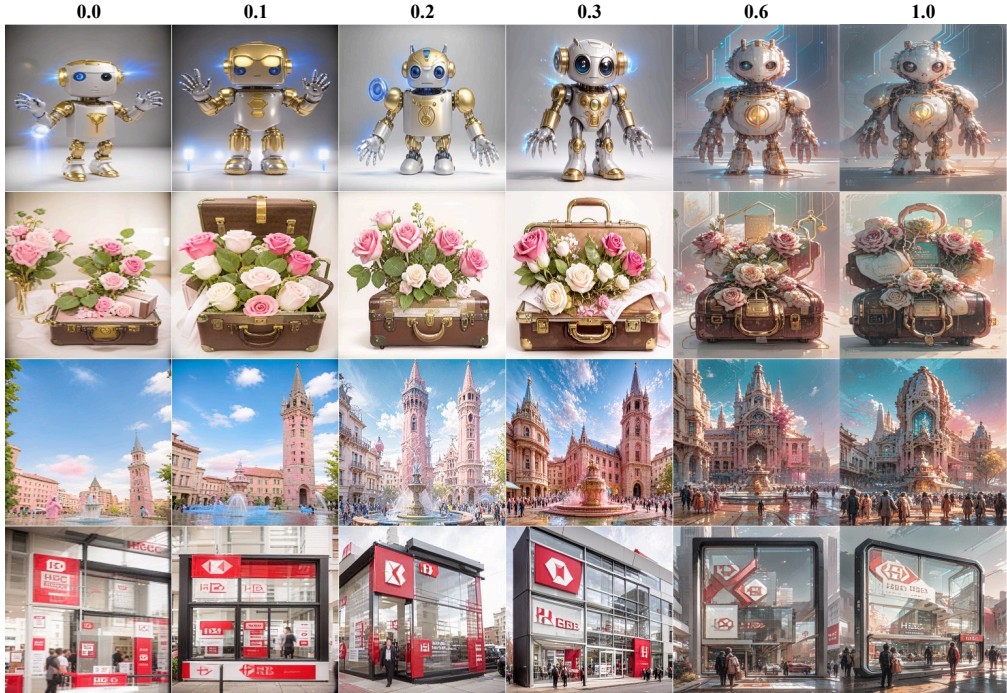

Figure 17: The generation results with and without our reweighting strategies at various reweighting ratios. When ratio is 1, it means the original loss.

## C.3 MORE EVALUATION

Here, we conduct additional evaluations across various structures and prompt lengths.

For structural variation, we use DPG-Bench (Hu et al., 2024), which includes test prompts for categories such as entity, attribute, relation, and count. The results in Table 4 show that our longSD model outperforms others.

For prompt length, we evaluate generation results for prompt lengths of about $N$ tokens, where $N \in [15, 60, 120, 240, 500]$. Specifically, for shorter prompts ($N \leq 240$), we use our test dataset and restrict the length to a maximum of $N$ tokens. If a sentence is truncated, it is discarded entirely. For a prompt is formatted as "xxx.xxx.xxx(N)xxx" and the N token appears in the middle of a sentence, we only retain "xxx.xxx.". For longer prompts around $N = 500$ tokens, we utilize GPT-4o for generation. During the T2I generation, the baseline models may truncate the prompt if the input exceeds their maximum limit, whereas the two evaluation methods still assess the generated images based on the entire given prompt. The evaluation results in Table 5 demonstrate that our method consistently surpasses current baselines.

Both additional evaluations highlight the effectiveness of our segment-level encoding and preference optimization strategies.

Table 4: Evaluation results on DPG-Bench of different models.

| Model | Average | Global | Entity | Attribute | Relation | Other |
|---|---|---|---|---|---|---|
| SD-2 | 68.09 | 77.67 | 78.13 | 74.91 | 80.72 | 80.66 |
| PlayG-2 | **74.54** | 83.61 | 79.91 | 82.67 | 80.62 | 81.22 |
| PixArt-$\alpha$ | 71.11 | 74.97 | 79.32 | 78.60 | 82.57 | 76.96 |
| KanD-2.2 | 70.12 | 77.07 | 80.01 | 77.55 | 80.94 | 78.64 |
| SD-1.5 | 63.18 | 74.63 | 74.23 | 75.39 | 73.49 | 67.81 |
| ELLA$_{1.5}$ | 74.91 | 84.03 | 84.61 | 83.48 | 84.03 | 80.79 |
| **longSD** | **77.58** | 76.27 | 83.00 | 86.40 | 86.52 | 86.21 |

Table 5: Denscore-O and VQAscore results for $512 \times 512$ image generation using different models and different maximum prompt lengths.

| Token | Metric | SD-1.5 | SD-2.1 | PlayG-2 | PixArt-$\alpha$ | KanD-2.2 | ELLA | longSD(S) | longSD(S+R) |
|---|---|---|---|---|---|---|---|---|---|
| 15 | Denscore-O | 25.31 | 26.51 | 23.90 | 27.84 | 29.04 | 27.13 | 26.36 | 29.07/**30.12** |
|  | VQAscore | 88.32 | 90.27 | 88.32 | 91.12 | 91.88 | 90.28 | 90.71 | 92.25/**92.52** |
| 60 | Denscore-O | 30.42 | 31.93 | 30.23 | 34.01 | 35.13 | 33.68 | 33.08 | 36.48/**37.14** |
|  | VQAscore | 84.54 | 87.04 | 86.13 | 88.73 | 87.93 | 87.98 | 87.90 | 89.03/**89.29** |
| 120 | Denscore-O | 29.49 | 30.59 | 29.26 | 33.72 | 33.75 | 33.65 | 31.71 | 34.52/**35.57** |
|  | VQAscore | 83.63 | 85.33 | 84.78 | 86.81 | 86.02 | 86.93 | 85.88 | 87.25/**87.49** |
| 240 | Denscore-O | 29.26 | 30.10 | 28.8 | 33.48 | 33.18 | 32.94 | 31.20 | 34.29/**35.18** |
|  | VQAscore | 84.69 | 85.81 | 85.38 | 87.19 | 86.58 | 86.90 | 86.27 | 87.26/**87.45** |
| 500 | Denscore-O | 15.11 | 15.12 | 13.22 | 16.27 | 16.78 | 16.83 | 15.90 | 19.01/**19.36** |
|  | VQAscore | 81.14 | 82.69 | 80.42 | 84.79 | 84.94 | 85.84 | 84.34 | 86.65/**87.24** |

## C.4 HIGH-RESOLUTION GENERATION

To evaluate our method for high-resolution generation with the latest models, we conducted experiments using SDXL (Podell et al., 2023) at a resolution of $1024 \times 1024$. We assess the results with FID, Denscore, VQAscore, and GPT4o, comparing them to those from the previously presented SD1.5 version. The results are shown in Table 10.

According to the results, we can see that (1) our methods significantly improve both SD1.5 and SDXL, demonstrating their robustness. (2) Among the two final fine-tuned versions, longSDXL clearly outperforms longSD1.5 in terms of long text alignment, indicating that a stronger foundation model achieves better performance limits. (3) The improvement is more pronounced in SD1.5, this is because the pretrained version of SDXL is already better than that of SD1.5.

In the following tables, we also provide visual results using SDXL:

Table 6: Comparison of Image Outputs with and without LongAlign (1/6)

| Image 1 (w/o LongAlign) | Image 2 (w LongAlign) |
|---|---|
| 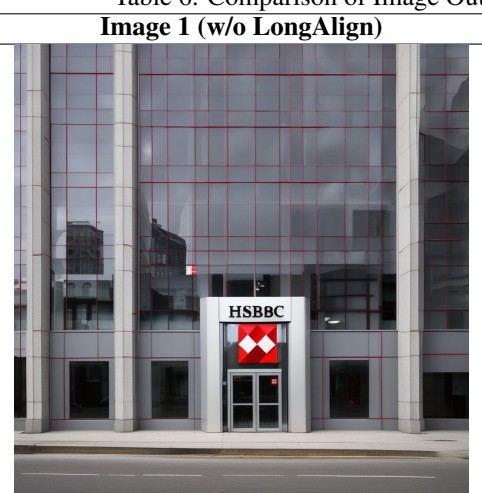 | 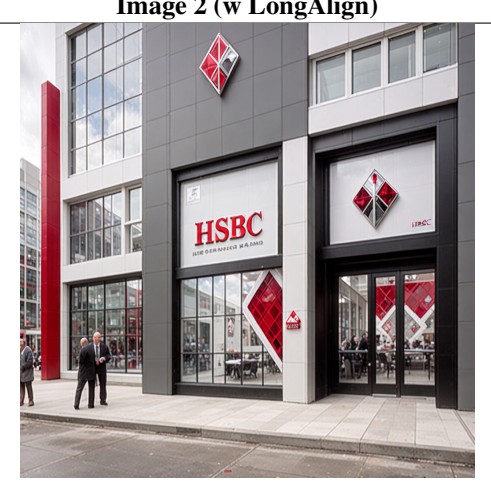 |

The image captures the exterior of an HSBC bank branch. Dominating the scene is a gray building, its facade punctuated by a large window on the right side. This window, framed in black, is divided into six panes, each reflecting the world outside. Above this window, a red and white sign proudly displays the HSBC logo. The letters "HSBC" are written in black, standing out against the white background of the sign. To the right of the logo, a red diamond-shaped symbol adds a splash of color to the scene. The image is a blend of urban architecture and corporate branding, a snapshot of a moment in the life of the city.

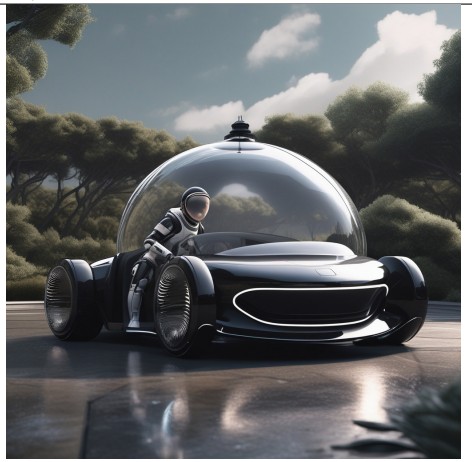 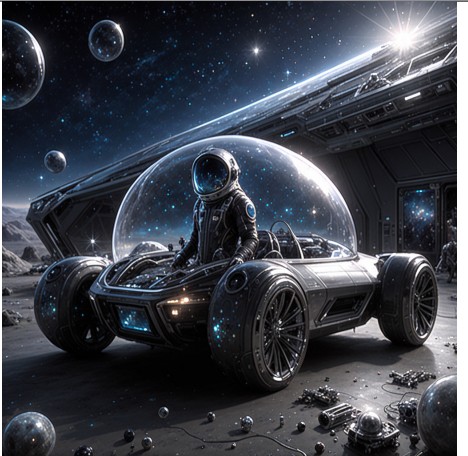

The image presents a 3D rendering of a futuristic car, which is the central focus of the composition. The car is predominantly black, with a glossy finish that reflects the surrounding environment. It's equipped with a large, transparent bubble-like dome on the roof, through which a small astronaut can be seen. The astronaut, dressed in a black suit with a helmet, is floating in space, surrounded by stars and planets. The car is not just a vehicle but a spacecraft, as indicated by the presence of the astronaut and the celestial backdrop. The car's design is sleek and modern, with a curved front and a pointed rear. The wheels are large and silver, adding to the futuristic aesthetic. The background of the image is a dark blue, providing a stark contrast to the black car and the astronaut. This contrast further emphasizes the car and the astronaut, making them the focal points of the image. Overall, the image is a blend of science fiction and modern design, creating a visually striking and imaginative scene.

Table 7: Comparison of Image Outputs with and without LongAlign (2/6)

| Image 1 (w/o LongAlign) | Image 2 (w LongAlign) |
|---|---|

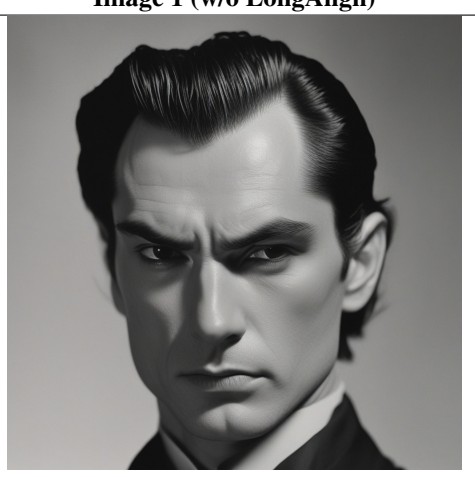 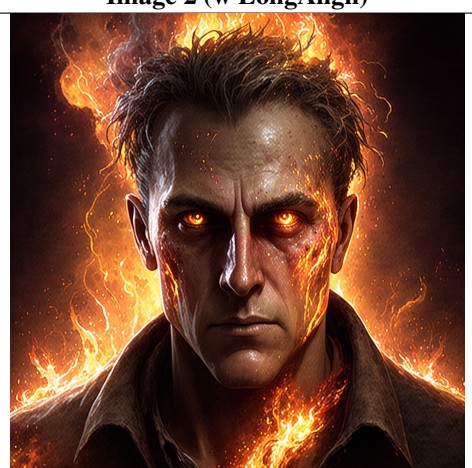

The image is a close-up portrait of a man with a serious expression. His hair is styled in a slicked-back manner, and his facial features are highlighted by the lighting, which casts a dramatic shadow on his face. The man's eyes are directed towards the camera, and his eyebrows are slightly furrowed, adding to the intensity of his expression. The most striking element of the image is the fire that appears to be emanating from the man's neck and shoulders. The fire is depicted in a realistic style, with orange and yellow hues that suggest a bright, intense flame. The fire is not contained within the image; it seems to be flowing outward, creating a sense of movement and energy. The background of the image is dark, which serves to highlight the man and the fire. The darkness also helps to emphasize the contrast between the man's skin and the fiery elements in the image. Overall, the image is a powerful and dramatic portrait that combines realistic elements with artistic flair. The use of fire as a visual motif adds a layer of intrigue and mystery to the image, inviting the viewer to wonder about the story behind the scene.

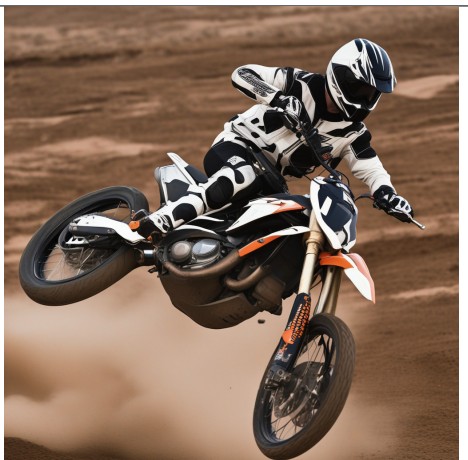 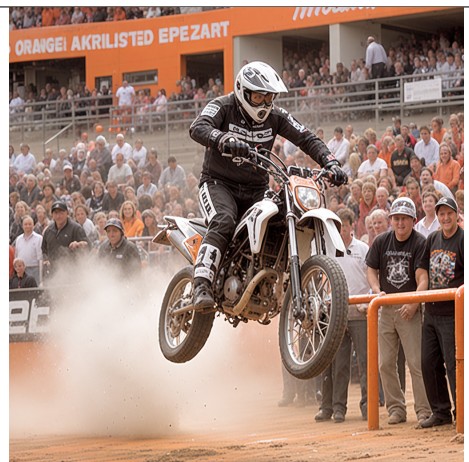

In the center of the image, a daring motorcyclist is captured in mid-air, performing a thrilling stunt on an orange and black dirt bike. The rider, clad in a black and white helmet, grips the handlebars tightly, demonstrating control and precision. The bike is tilted slightly to the left, adding to the sense of motion and excitement. The setting is a large indoor stadium, filled with a crowd of spectators who are watching the spectacle unfold. Their faces are a blur of anticipation and awe. The background is adorned with various advertisements, adding a splash of color and life to the scene. Despite the action-packed nature of the image, there's a certain harmony to it. The motorcyclist, the bike, the crowd, and the stadium all come together to create a snapshot of a moment filled with adrenaline and excitement. It's a testament to the skill and courage of the rider, and the thrill of the sport.

Table 8: Comparison of Image Outputs with and without LongAlign (4/6)

| Image 1 (w/o LongAlign) | Image 2 (w LongAlign) |
|---|---|
| 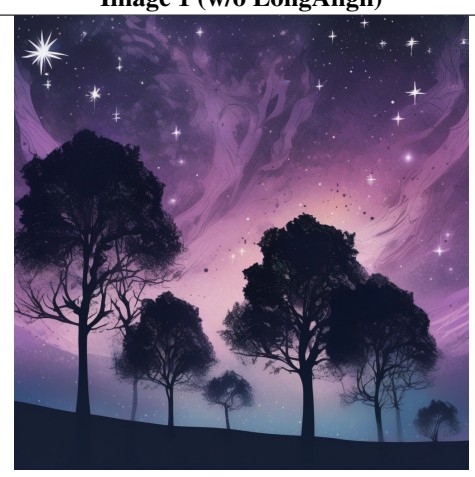 | 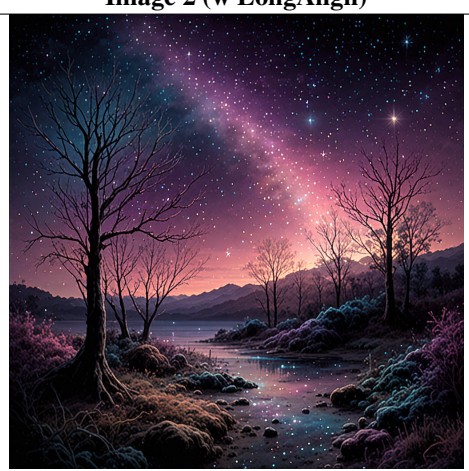 |

The image is a digital artwork depicting a nighttime scene. In the foreground, there are three silhouetted trees with curved branches, suggesting a tranquil setting. The trees are set against a dark sky, which is filled with numerous stars and a faint, milky-way-like nebula. The colors in the sky transition from deep blues at the top to lighter purples and pinks near the horizon, indicating either dawn or dusk. The overall atmosphere of the image is serene and somewhat mystical, with a sense of depth and vastness conveyed by the starry sky. There are no visible texts or distinguishing marks that provide additional context or information about the image. The style of the artwork is realistic with a focus on creating a peaceful and somewhat ethereal nighttime landscape.

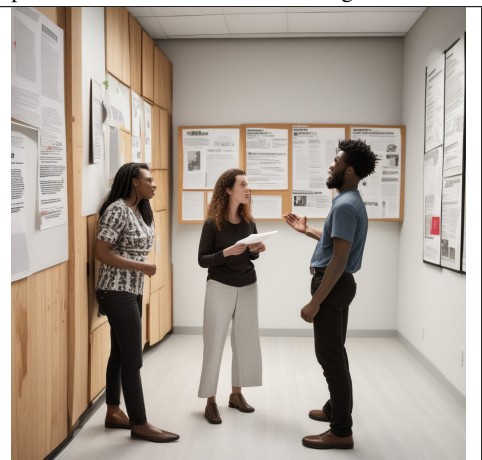 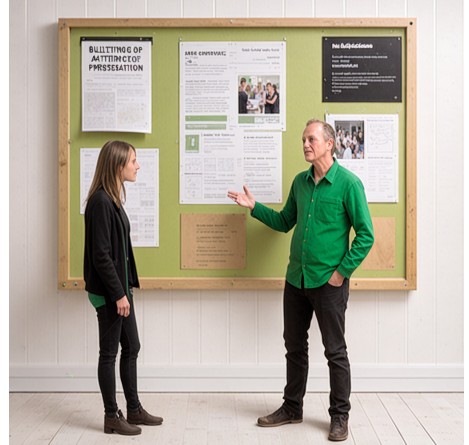

In the image, there's a lively scene unfolding in a room with a wooden floor and a white wall in the background. Two individuals are engaged in a discussion, standing in front of a bulletin board. The bulletin board, made of wood, is adorned with two posters. One poster is white with black text, while the other is black with white text. The person on the left, clad in a green shirt, is gesturing with their hands, perhaps emphasizing a point or explaining something. On the right, the other person, wearing a black jacket, is attentively listening, their gaze fixed on the person in green. The interaction between the two individuals suggests a discussion or presentation of some sort. The bulletin board, with its two posters, serves as the backdrop for this exchange.

Table 9: Comparison of Image Outputs with and without LongAlign (6/6)

| Image 1 (w/o LongAlign) | Image 2 (w LongAlign) |
|---|---|
| 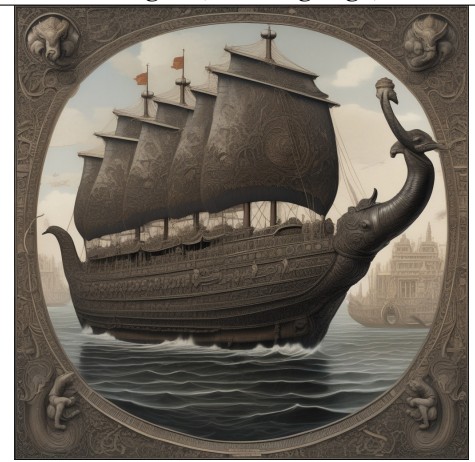 | 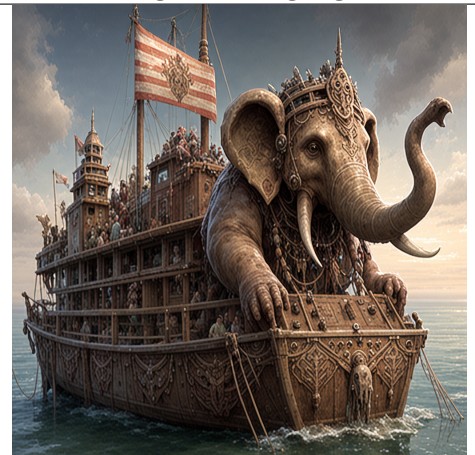 |

The image depicts a fantastical scene featuring a large ship with a distinctive design. The ship is predominantly dark in color, with intricate carvings and decorations that suggest a historical or mythical inspiration. The most striking feature of the ship is the presence of a large, elephant-like head at the bow, which is facing towards the viewer. This head is detailed and realistic, with tusks and a trunk that are prominently displayed. The ship is sailing on a body of water, with a clear sky above and a calm, reflective surface below. In the background, there is a rocky outcrop that adds to the sense of a natural, outdoor setting. On the deck of the ship, there are several people visible, although they are too small to discern any specific details about them. The ship is also adorned with multiple flags, which are attached to the mast and the bow. These flags are not detailed enough to identify any specific symbols or insignias. The overall style of the image is realistic with a touch of fantasy, as evidenced by the elephant head and the elaborate carvings on the ship. The lighting and shadows suggest that the image is set during the daytime, with the sun casting a warm glow on the scene. There are no visible texts or brands in the image.

| | |
|---|---|
| 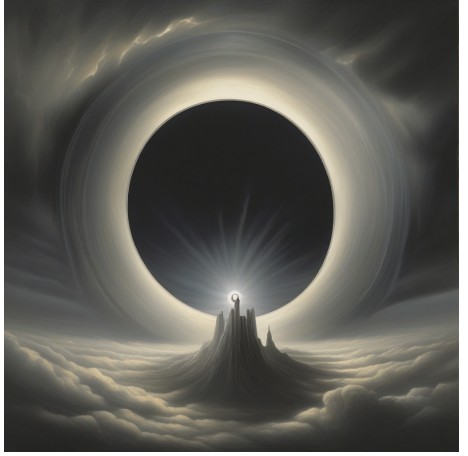 | 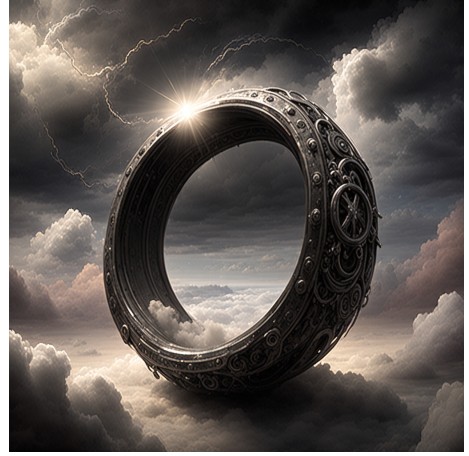 |

The image presents a dramatic and surreal scene set against a dark, cloudy sky. Dominating the center of the image is a large, black ring, which appears to be a portal or vortex. This ring is encircled by a halo of light, creating a stark contrast against the dark sky. The light seems to emanate from the center of the ring, suggesting a source of power or energy. Beyond the ring, the sky is filled with clouds that are illuminated by the light from the ring. The clouds are dense and appear to be in motion, adding a sense of dynamism to the scene. The colors in the image are predominantly dark and black, with the light from the ring providing a stark contrast. The overall composition of the image is balanced, with the ring centrally located and the clouds filling the rest of the frame. The use of light and shadow creates a sense of depth and dimension, drawing the viewer's eye towards the center of the image. Despite the fantastical elements, the image is grounded in a realistic aesthetic. The clouds and sky are rendered with a high level of detail, and the colors are naturalistic. This combination of realism and fantasy creates a visually striking image that invites the viewer to imagine what lies beyond the ring.

Table 10: Evaluation results of our methods on two foundation models: SD1.5 and SDXL.

| Model | FID | Denscore-O | Denscore | VQAscore | GPT-4o |
|-------|-----|-----------|----------|----------|--------|
| SD1.5 | 24.96 | 29.29 | 20.29 | 84.57 | 195 |
| longSD1.5 | 24.28 | 35.26 | 23.79 | 87.24 | 668 |
| SDXL | 21.18 | 33.52 | 22.79 | 86.89 | 268 |
| longSDXL | 23.88 | 37.33 | 25.33 | 87.30 | 416 |

## D  GPT-4O EVALUATION

The input for evaluation consists of a prompt and two images generated for this prompt using different models. We utilize the multimodal evaluation capabilities of GPT-4o and employ the following Python code for evaluation:

```python
def eval_fn(prompt, image1, image2):
    template = f""" {prompt}
    Which image is more consistent with the above prompt? Please respond
        with "first", "second" or "tie", no explanation needed. """
    completion = client.chat.completions.create(
        model="gpt-4o-2024-05-13",
        messages=[{
                "role": "system", "content": "You are an image generation
                    evaluation expert.",
                "role": "user", "content": [
                    {"type": "text", "text": template},
                    {"type": "image_url", "image_url": {"url": f"data:
                        image/jpeg;base64,{image1}"}},
                    {"type": "image_url", "image_url": {"url": f"data:
                        image/jpeg;base64,{image2}"}},
            ]}])
```

Table 11: Evaluation for comparison in the P2I diffusion framework.

| Method | 768 | | 1024 | |
|--------|-----|------|------|------|
| | P2I | +ours | P2I | +ours |
| FID-5k | 20.36 | 21.60 | 19.78 | 20.84 |
| Denscore-O | 34.45 | 38.71 | 34.78 | 38.51 |
| Denscore | 23.43 | 25.39 | 23.47 | 25.41 |
| GPT-4o | 240 | 583 | 289 | 536 |

## E  OUT-OF-DISTRIBUTION PROBLEM

Some of these approaches incorporate the assistance of LLMs, such as Ranni and RPG-Diffusers. LLM-based methods that involve additional LLM assistance increase computational requirements and encounter significant out-of-distribution (OOD) issues with long-text inputs.

There are several reasons why existing LLM-based methods struggle with long-text inputs. Long texts contain more facts than shorter ones, and LLMs rely on injecting different subprompts into specific subareas. Some facts may be disjoint, while others overlap (e.g., detailed descriptions of a single object), complicating the separation of information in pixel space. Additionally, some LLM-based methods fine-tune the LLM model that may not include datasets with long and detailed inputs. As a result, these models often cannot manage such situations for generating effective layout plans to assign different subprompts across different areas.

While current LLM-based methods are not fully effective, we believe this OOD problem can be resolved in the future. Additionally, LLM-based methods complement our approach: while we concentrate on training more powerful foundation models, these techniques can further enhance results during the sampling stage.

Here are some examples of OOD problems:

**Ranni**. In our experiment with Ranni, approximately half of the prompts resulted in errors before generating the final output. We selected examples from the remaining successful prompts. The example prompt is generated from the first image in Figure 18. Although Ranni generates the layout initially, it creates too many subareas that do not contribute clearly to the final result, as shown in the second image in Figure 18.

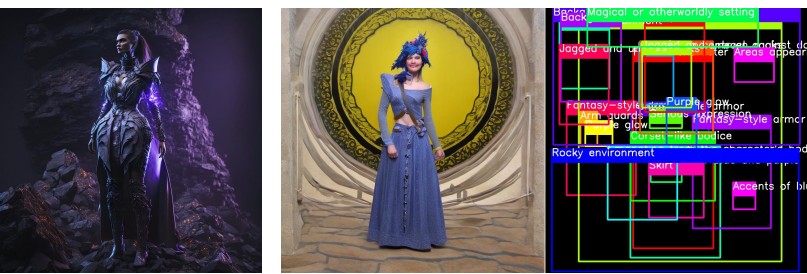

Figure 18: The reference image and the corresponding generated image using Ranni.

The complete input prompt is "The image is a digital artwork featuring a female character standing in a rocky environment. The character is dressed in a fantasy-style armor with a predominantly dark color scheme, highlighted by accents of blue and purple. The armor includes a corset-like bodice, a skirt, and arm guards, all adorned with intricate designs and patterns. The character's hair is styled in a high ponytail, and she has a serious expression on her face. The armor is illuminated by a purple glow, which appears to emanate from the character's body, creating a contrast against the darker elements of the armor and the surrounding environment. The glow also casts a soft light on the character's face and the armor, enhancing its details and textures. The background consists of a rocky landscape with a purple hue, suggesting a magical or otherworldly setting. The rocks are jagged and uneven, with some areas appearing to be on fire, adding to the dramatic and intense atmosphere of the image. There are no visible texts or logos in the image, and the style of the artwork is realistic with a focus on fantasy elements. The image is likely intended for a gaming or fantasy-themed context, given the character's attire and the overall aesthetic."

**RPG-Diffusers**. For RPG-Diffusers, the example prompt is generated from the first image in Figure 19. However, they assign two subareas: the left side and the right side of the image. Using these subprompts—"The black leather sofa is stylishly centered in the frame, showcasing its plush texture and elegant design, the distinct cushions maintain the streamlined look of the piece, offering comfort against the stark contrast of the white background." and "The three luxurious black cushions are artfully arranged, emphasizing the sofa's inviting nature while seamlessly blending into the cohesive monochrome theme of the setting."—results in a repetition of "soft" in the generated image, as shown in the second image in Figure 19.

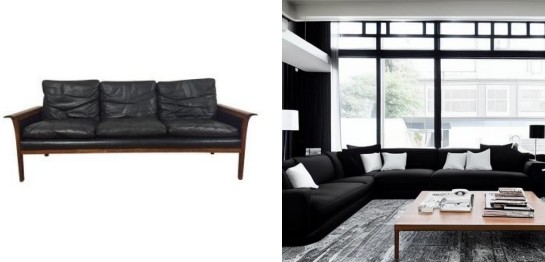

Figure 19: The reference image and the corresponding generated image using RPG-Diffusers.

The complete input prompt is "The image presents a scene dominated by a black leather sofa, which is the central object in the frame. The sofa, with its three cushions, is positioned against a stark white background, creating a striking contrast. The cushions, like the sofa, are black and appear to be made of leather, suggesting a uniform color scheme throughout the piece. The sofa is designed

with a wooden frame, adding a touch of warmth to the otherwise monochrome setting. The frame is visible on both the front and back of the sofa, providing stability and support. The arms of the sofa, like the frame, are made of wood, maintaining the overall aesthetic of the piece. The sofa is set against a white background, which accentuates its black color and wooden frame, making it the focal point of the image. There are no other objects in the image, and no text is present. The relative position of the sofa is central, with ample space around it, further emphasizing its importance in the image. The image does not depict any actions, but the stillness of the sofa suggests a sense of tranquility."

## F    TEXT CONDITION FOR VISUAL RESULT

In this section, we provide the text conditions for the visual results in Figure 1 and Figure 6. The text conditions are as follows:

The text conditions for Figure 1 are as follows:

1. The image presents a 3D rendering of a horse, captured in a profile view. The horse is depicted in a state of motion, with its mane and tail flowing behind it. The horse's body is composed of a network of lines and curves, suggesting a complex mechanical structure. This intricate design is further emphasized by the presence of gears and other mechanical components, which are integrated into the horse's body. The background of the image is a dark blue, providing a stark contrast to the horse and its mechanical components. The overall composition of the image suggests a blend of organic and mechanical elements, creating a unique and intriguing visual.

2. The image presents a close-up view of a human eye, which is the central focus. The eye is surrounded by a vibrant array of flowers, predominantly in shades of blue and purple. These flowers are arranged in a semi-circle around the eye, creating a sense of depth and perspective. The background of the image is a dark blue sky, which contrasts with the bright colors of the flowers and the eye itself. The overall composition of the image suggests a theme of nature and beauty.

3. The image presents a detailed illustration of a submarine, which is the central focus of the artwork. The submarine is depicted in a three-quarter view, with its bow facing towards the right side of the image. The submarine is constructed from wood, giving it a rustic and aged appearance. It features a dome-shaped conning tower, which is a common feature on submarines, and a large propeller at the front. The submarine is not alone in the image. It is surrounded by a variety of sea creatures, including fish and sharks, which are swimming around it. These creatures add a sense of life and movement to the otherwise static image of the submarine. The background of the image is a light beige color, which provides a neutral backdrop that allows the submarine and the sea creatures to stand out. However, the background is not devoid of detail. It is adorned with various lines and text, which appear to be a map or a chart of some sort. This adds an element of intrigue to the image, suggesting that the submarine might be on a mission or an expedition. Overall, the image is a detailed and intricate piece of art that captures the essence of a submarine voyage, complete with the submarine, the sea creatures, and the map in the background. It's a snapshot of a moment in time, frozen in the image, inviting the viewer to imagine the stories and adventures that might be taking place beneath the surface of the water.

4. In the image, there's a charming scene featuring a green frog figurine. The frog, with its body painted in a vibrant shade of green, is the main subject of the image. It's wearing a straw hat, adding a touch of whimsy to its appearance. The frog is positioned in front of a white window, which is adorned with a green plant, creating a harmonious color palette with the frog's body. The frog appears to be looking directly at the camera, giving the impression of a friendly encounter. The overall image exudes a sense of tranquility and simplicity.

5. The image portrays a female character with a fantasy-inspired design. She has long, dark hair that cascades down her shoulders. Her skin is pale, and her eyes are a striking shade of blue. The character's face is adorned with intricate gold and pink makeup, which includes elaborate patterns and designs around her eyes and on her cheeks. Atop her head, she wears a crown made of gold and pink roses, with the roses arranged in a circular pattern. The crown is detailed, with each rose appearing to have a glossy finish. The character's attire consists of a gold and pink dress that is embellished with what appears to be feathers or leaves, adding to the fantasy aesthetic. The

background of the image is dark, which contrasts with the character's pale skin and the bright colors of her makeup and attire. The lighting in the image highlights the character's features and the details of her makeup and attire, creating a dramatic and captivating effect. There are no visible texts or brands in the image. The style of the image is highly stylized and artistic, with a focus on the character's beauty and the intricate details of her makeup and attire. The image is likely a digital artwork or a concept illustration, given the level of detail and the fantastical elements present.

6. The image captures a scene of a large, modern building perched on a cliff. The building, painted in shades of blue and gray, stands out against the backdrop of a cloudy sky. The cliff itself is a mix of dirt and grass, adding a touch of nature to the otherwise man-made structure. In the foreground, a group of people can be seen walking along a path that leads up to the building. Their presence adds a sense of scale to the image, highlighting the grandeur of the building. The sky above is filled with clouds, casting a soft, diffused light over the scene. This light enhances the colors of the building and the surrounding landscape, creating a visually striking image. Overall, the image presents a harmonious blend of architecture and nature, with the modern building seamlessly integrated into the natural landscape.

The text conditions for Figure 6 are as follows:

1. The image is a digital artwork featuring a female character standing in a rocky environment. The character is dressed in a fantasy-style armor with a predominantly dark color scheme, highlighted by accents of blue and purple. The armor includes a corset-like bodice, a skirt, and arm guards, all adorned with intricate designs and patterns. The character's hair is styled in a high ponytail, and she has a serious expression on her face. The armor is illuminated by a purple glow, which appears to emanate from the character's body, creating a contrast against the darker elements of the armor and the surrounding environment. The glow also casts a soft light on the character's face and the armor, enhancing its details and textures. The background consists of a rocky landscape with a purple hue, suggesting a magical or otherworldly setting. The rocks are jagged and uneven, with some areas appearing to be on fire, adding to the dramatic and intense atmosphere of the image. There are no visible texts or logos in the image, and the style of the artwork is realistic with a focus on fantasy elements. The image is likely intended for a gaming or fantasy-themed context, given the character's attire and the overall aesthetic.

2. The image presents a scene of elegance and luxury. Dominating the center of the image is a brown Louis Vuitton suitcase, standing upright. The suitcase is adorned with a gold handle and a gold lock, adding a touch of opulence to its appearance. Emerging from the top of the suitcase is a bouquet of pink and white roses, interspersed with green leaves. The roses, in full bloom, seem to be spilling out of the suitcase, creating a sense of abundance and luxury. The entire scene is set against a white background, which accentuates the colors of the suitcase and the roses. The image does not contain any text or other discernible objects. The relative position of the objects is such that the suitcase is in the center, with the bouquet of roses emerging from its top.

3. The image captures the grandeur of the Toledo Town Hall, a renowned landmark in Toledo, Spain. The building, constructed from stone, stands tall with two prominent towers on either side. Each tower is adorned with a spire, adding to the overall majesty of the structure. The facade of the building is punctuated by numerous windows and arches, hinting at the intricate architectural details within. In the foreground, a pink fountain adds a splash of color to the scene. A few people can be seen walking around the fountain, their figures small in comparison to the imposing structure of the town hall. The sky above is a clear blue, providing a beautiful backdrop to the scene. The image is taken from a low angle, which emphasizes the height of the town hall and gives the viewer a sense of being in the scene. The perspective also allows for a detailed view of the building and its surroundings. The image does not contain any discernible text. The relative positions of the objects confirm that the town hall is the central focus of the image, with the fountain and the people providing context to its location.

