# OpenReview forum: "Improving Long-Text Alignment for Text-to-Image Diffusion Models"
_ICLR.cc/2025/Conference — ICLR 2025 Poster_

### Official Review · Reviewer_Hfc6 · 2024-11-03

**Soundness:** 3
**Presentation:** 3
**Contribution:** 2
**Rating:** 6
**Confidence:** 4

**Summary:**

This paper presents a new approach for long text inputs on text-to-image (T2I) alignment since the clip text encoding only allows 77 tokens. The authors address limitations of existing encoding methods like CLIP by proposing segment-level encoding, where long texts are divided and processed in parts to bypass input length constraints. They further introduce a decomposed preference optimization that separates alignment-related and non-alignment components of CLIP-based preference scores. By reweighting these components, the method reduces overfitting, achieving superior T2I alignment after fine-tuning Stable Diffusion v1.5, outperforming models such as PixArt-α and Kandinsky v2.2.

**Strengths:**

(1) Comprehensive Survey of Related Work.
The paper presents a thorough and comprehensive survey of existing work in text-to-image (T2I) diffusion models, demonstrating an impressive grasp of the field. By delving deeply into previous approaches and their limitations, the authors effectively set the stage for their contributions, clarifying the gaps their method aims to fill. This background provides readers with valuable context and insight into the evolution of T2I models, particularly in handling longer, complex textual inputs. The comprehensive nature of this survey also reinforces the authors' understanding of the field's current challenges and strengths, building confidence in the relevance and timeliness of the proposed approach.

(2) Importance of the Problem and a Reasonable, Well-Motivated Solution.
The authors tackle a critical issue in T2I diffusion models: the difficulty of aligning generated images with longer text prompts. As the demand for complex, high-fidelity image generation grows, the ability to handle longer text inputs accurately is essential. The segmentation approach, paired with decomposed preference optimization, offers a well-motivated solution to this problem. Segmenting long text into manageable parts allows for better processing within the confines of existing encoding models, while the decomposed preference optimization fine-tunes the alignment, addressing the unique challenges posed by long prompts. The design choices reflect a reasonable and methodical approach to tackling these limitations, and the paper articulates the rationale for each component clearly. This structured approach suggests the authors have carefully considered the problem’s nuances, offering a solution that is not only effective but also grounded in sound methodology.

(3) Demonstrated Superiority over State-of-the-Art Models.
One of the paper's significant strengths is the demonstrated performance improvement over state-of-the-art models. Through rigorous experimentation, the authors show that their method surpasses current leading models like PixArt-α and Kandinsky v2.2 in T2I alignment, particularly for long-text prompts. By fine-tuning Stable Diffusion v1.5 with their approach, they achieve superior alignment, reducing overfitting while preserving text-relevant information in the generated images. This achievement underscores the potential of the proposed method to set a new benchmark for handling longer, more detailed textual inputs within T2I models. The improvement over established models validates the effectiveness of the segmentation and preference optimization strategy, indicating that this approach could meaningfully advance the state of the art in T2I diffusion modeling.

**Weaknesses:**

Although the proposed method solved an important issue, three major issues remain as listed below.
(1) Limitations and Ambiguities in the Segmentation and Merging Methodology. The segmentation and merging technique proposed in this work introduces a unique approach to handling longer text inputs but raises questions regarding its effectiveness and generalizability. When text inputs exceed 77 tokens, this method still encounters limitations, as it is fundamentally restricted by the underlying model’s capacity to handle “long” sequences since the split-and-merge process does not solve the problem. This constraint is particularly concerning as longer text inputs are common in real-world applications and often essential to producing detailed and contextually accurate image generations. The current approach of segmenting and then merging these sections seems like a workaround rather than a robust solution to handling extended texts, which may inherently limit its scalability and versatility. Furthermore, the mechanics of how segmentation and merging affect the underlying model's cross-attention dynamics remain underexplored. Cross-attention is a critical component in the alignment process between text and image features, and segmenting inputs may disrupt this alignment, especially as certain semantic connections might be lost or diluted across segmented inputs. Investigating the cross-attention differences between the original, unsegmented approach and the segment-and-merge methodology could shed light on any distortions introduced by this technique. A more thorough analysis of cross-attention’s role here could help refine segmentation methods to better retain textual coherence and improve image alignment fidelity, ultimately benefiting downstream performance.

(2) Dependency on an Outdated Baseline Model (Stable Diffusion v1.5):
The use of Stable Diffusion v1.5 as the primary evaluation model poses a significant limitation, given that the field has moved toward more advanced versions like SD-3 and SDXL. These newer versions incorporate improved architectures and training techniques, yielding enhanced performance, especially in terms of image quality and alignment with textual inputs. The reliance on an outdated model not only limits the relevance of the study’s results but also restricts the potential impact of the proposed method. Using v1.5 as the baseline reflects well on the approach’s applicability to older architectures, but it leaves unanswered questions about its efficacy on more sophisticated models that incorporate advancements in diffusion techniques, training scale, and multimodal alignment mechanisms.
Moreover, maintaining SD-1.5 as a standard for comparison could inadvertently hold back progress within the research community. As models continue to evolve, it’s essential to align benchmark tests with the latest technologies to ensure that methods are relevant and that advancements reflect real-world capabilities. Preliminary results from newer models, such as SD-3, have demonstrated considerable improvements in T2I alignment, indicating that the proposed method may benefit even further from these architectural updates. Testing on newer models would better position the approach in the context of current technological standards, ensuring that it remains relevant and applicable as diffusion models evolve. Future work should include evaluations on SD-3 and SDXL to substantiate claims of superiority over other methods in a more current setting. The test of SD-3 with the prompt used in the first example of Fig. 1 is shown below.
https://ibb.co/CWyKQTZ

(3) Over-reliance on Long Prompt Training and Lack of Generalizability Testing.
The proposed method seems to rely heavily on training with long prompts, which could limit its flexibility and adaptability. While training on extended text inputs may enhance alignment for similar prompts, it raises concerns about the model's performance on shorter or more varied prompts. In real-world scenarios, prompt lengths and structures vary significantly, and a robust model should perform consistently across this spectrum. By focusing predominantly on long-prompt alignment, the current approach may overfit to specific input lengths, making it less effective for shorter or less detailed prompts where segmentation might not be necessary or where text segments are not sufficiently complex to benefit from this treatment.
To address this potential limitation, it would be valuable to conduct experiments that vary prompt lengths and structures systematically, assessing whether the model’s performance holds across different scenarios. Additionally, testing with alternative segmentation designs could reveal whether simpler or more complex methods yield better alignment. These experiments would enhance our understanding of how adaptable the proposed method is, providing insights into its generalizability and robustness. The community would benefit from such insights, as they could guide further development of segmentation-based approaches for T2I tasks.

**Questions:**

(1) Please better explain why the proposed split-and-merge approach can address the long-text alignment issue.
(2) Please provide the ablation study clearly.

---

> ### Author Response · Authors · 2024-11-20
> **Rebuttal by Authors [1/2]**
>
> Thank you for your constructive feedback. Your insightful questions have improved our work very much. Below we respond to the comments in weaknesses (***W***) and questions (***Q***).
>
> ---
>
> ***W1: These sections seem like a workaround rather than a robust solution to handling extended texts.***
>
> We agree that segment encoding alone cannot solve all long text encoding problems, but it can enhance any existing model with a maximum input limitation. In practice, as the total maximum length increases, data collection becomes increasingly challenging. For instance, generating 250-token prompts with LLAVA is straightforward, while collecting 1000-token prompts using existing models (e.g. LLMs) is significantly harder. Our method enables us to extend an encoder trained on a maximum of 250 tokens to accommodate 1000-token inputs. Additionally, a 1000-token prompt can often be divided into several relatively independent sections, each with fewer than 250 tokens, making segment encoding for these sections a practical and sensible choice.
>
> ---
>
> ***W1.2: The mechanics of how segmentation and merging affect the underlying model's cross-attention dynamics remain underexplored.***
>
> Thank you for your advice. In $\\textcolor{blue}{\\textrm{Figure 12}}$ of the new version, we compare the original SD1.5 with our fine-tuned version and find that their cross-attention map behaviors are similar, regardless of whether segment encoding is used. Specifically, when the prompt accurately labels each object and references them in subsequent sentences, even though segment encoding does not process multiple sentences in a single forward pass, the attention maps for individual objects across different segments remain consistent. For example, the prompt is structured as "A dog xxx and a cat xxx. This dog xxx." Even though the details about this dog are divided into two segments in our setup, the model can identify the same dog and apply attributes from both segments accordingly. This shows that T2I models with segment encoding can identify and manage information across segments of long-text inputs.
>
> ---
>
> ***W2: Future work should include evaluations on SD-3 and SDXL to substantiate claims of superiority over other methods in a more current setting.***
>
> We have added new foundation models, SDXL. Our method also extends the maximum input token limit for SDXL and improves long text alignment. All training and testing settings match the SD1.5 version, but with 1024 resolutions. The evaluation results on 5k test set are shown below:
>
> |  | FID | Denscore-O | Denscore | VAQscore | GPT4o |
> | :---- | :---- | :---- | :---- | :---- | :---- |
> | SDXL | 21.18 | 33.52 | 22.79 | 86.89 | 268 |
> | longSDXL | 23.88 | 37.33 | 25.33 | 87.30 | 416 |
>
> ---
>
> ***W3: It raises concerns about the model's performance on shorter or more varied prompts.***
>
> **Diversified Length:** In $\\textcolor{blue}{\\textrm{Table 6}}$ of our revised paper, we assess the generation results for prompts of lengths about 15, 60, 120, 240, and 500 tokens using Denscore-O and VQAscore \[1\]. For more details on the dataset construction and evaluation process, please see Appendix C.3 of the new version. The evaluation results in $\\textcolor{blue}{\\textrm{Table 6}}$ demonstrate that our method consistently outperforms current baselines.
>
> Here is a summary of the **VQAscore results** in $\\textcolor{blue}{\\textrm{Table 6}}$ in Appendix C.3:
>
> | Token | SD-1.5 | SD-2 | PlayG-2 | PixArt-$\\alpha$ | KanD-2.2 | ELLA-1.5 | LongSD (ours) |
> | :---- | :---- | :---- | :---- | :---- | :---- | :---- | :---- |
> | 15 | 88.32 | 90.27 | 88.32 | 91.12 | 91.88 | 90.28 | **92.52** |
> | 120 | 83.63 | 85.33 | 84.78 | 86.81 | 86.02 | 86.93 | **87.49** |
> | 500 | 81.14 | 82.69 | 80.42 | 84.79 | 84.94 | 85.84 | **87.24** |
>
> **Diversified structure:** For prompts with diversified structures, we test our model on DPG-Bench \[1\], which includes test prompts for various categories such as entity, attribute, relation, and count. The new results are shown in $\\textcolor{blue}{\\textrm{Table 5}}$ of Appendix C.3. Compared to the baselines, we have also made obvious improvements.
>
> Here is a summary of $\\textcolor{blue}{\\textrm{Table 5}}$ in Appendix C.3:
>
> | Model | SD-2 | PlayG-2 | PixArt-$\\alpha$ | KanD-2.2 | SD-1.5 | ELLA-1.5 | LongSD (ours) |
> | :---- | :---- | :---- | :---- | :---- | :---- | :---- | :---- |
> | DPG-Bench | 68.09 | 74.54 | 71.11 | 70.12 | 63.18 | 74.91 | **77.58** |
>
> \[1\] Hu X, Wang R, Fang Y, et al. Ella: Equip diffusion models with llm for enhanced semantic alignment\[J\]. arXiv preprint arXiv:2403.05135, 2024\.

---

> > ### Comment · Reviewer_Hfc6 · 2024-11-21
> > **Thank you for the rebuttal.**
> >
> > Thanks for the efforts on addressing the issues. I have raised the rating accordingly. Would you mind discussing the results on SDXL and compare the results with the results based on SD1.5.

---

> ### Author Response · Authors · 2024-11-20
> **Rebuttal by Authors [2/2]**
>
> ***W3.1: Testing with alternative segmentation designs could reveal whether simpler or more complex methods yield better alignment.***
>
> We examine different segmentation strategies by comparing the approach of treating each sentence as a segment versus grouping several consecutive sentences into one segment, as long as the total token count remains under 77\. Our new ablation study in $\\textcolor{blue}{\\textrm{Figure 11}}$ in the new version reveals no significant differences in the results.
>
> ---
>
> ***Q1: Please better explain why the proposed split-and-merge approach can address the long-text alignment issue.***
>
> We would like to clarify that the split-and-merge approach alone does not address long-text alignment. Segment encoding is a method to overcome the input limitations of the encoder. To enhance long-text alignment, we need an additional two-stage training process that includes both supervised fine-tuning and preference optimization.
>
> ---
>
> ***Q2: Please provide the ablation study clearly.***
>
> Thank you for your suggestion. In response, we have revised the paper to include new experimental results and detailed ablation studies, with all updates clearly highlighted in blue. We hope these additions address your concerns regarding the clarity of the ablation studies. If there are specific aspects you would like us to elaborate on further, we would be happy to provide additional results or analyses upon your guidance.

---

> ### Author Response · Authors · 2024-11-22
> **Thank you for your support**
>
> Thank you for raising the score\! The evaluation results for SD1.5 and SDXL are summarized as follows:
>
> |  | Denscore-O | Denscore | VAQscore | GPT4o |
> | :---- | :---- | :---- | :---- | :---- |
> | SD1.5 | 29.20 | 20.29 | 84.57 | 195 |
> | longSD1.5 | 35.26 (+6.06) | 23.79 (+3.50) | 87.24 (+2.67) | 668 (+473) |
> | SDXL | 33.52 | 22.79 | 86.89 | 268 |
> | longSDXL | 37.33 (+3.81) | 25.33 (+2.54) | 87.30 (+0.41) | 416 (+148) |
>
> According to these results, we observe that (1) our methods significantly improve both SD1.5 and SDXL, demonstrating their robustness. (2) Among the two final fine-tuned versions, longSDXL clearly outperforms longSD1.5 in terms of long text alignment, indicating that a stronger foundation model achieves better performance limits. (3) The improvement is more pronounced in SD1.5, this is because the pretrained version of SDXL is already better than that of SD1.5. We have uploaded a new revision that includes these analyses in Appendix C.4, with changes highlighted in blue for clarity.

---

### Official Review · Reviewer_hkae · 2024-11-03

**Soundness:** 4
**Presentation:** 2
**Contribution:** 3
**Rating:** 6
**Confidence:** 4

**Summary:**

This paper presents a method for enhancing the prompt following of text-to-image models specifically in the case of long prompts. The key contribution for tackling this problem is twofold: a) using a combination of CLIP and T5 encoders (as is becoming increasingly common these days e.g. SD3, Flux) b) the introduction of a preference model tailored for long prompts (Denscore) and applying reward fine-tuning with this Denscore model to enhance the prompt following of SD1.5 models for long prompts.

**Strengths:**

The paper tackles the crucial challenge of long prompt following in a very effective manner. Using a text encoder that can take the entire long prompt is a sound idea, and the Denscore preference model looks like a useful contribution in general.
Apart from this, the reward fine-tuning with the orthogonal decomposition and the gradient reweighting looks like a good idea to deal with the "reward-hacking" problem.
Finally, the results also appear quite strong from the evaluations presented in the paper.

**Weaknesses:**

An important paper that is missed here is ELLA[Hu et al. 2024] for a couple of reasons. The first is that they propose replacing the CLIP encoder of SD1.5 with a T5-XL model and get significantly improved results (far superior numbers to those reported by Lavi-Bridge whose MLP adapter is used here). Therefore, this model might be a valid comparison (although the training cost of ELLA is a bit higher: 7 days with 8 A100s for SD1.5). Alternatively, the adapter provided by ELLA would have probably been a better alternative to the one used in the paper (from Lavi-Bridge).

Apart from the comparison/use of adapter, there's also DPG-Bench introduced in the paper which is a good benchmark for long prompt following (as compared to existing benchmarks like T2I-Compbench, DSG, TIFA etc.). Evaluating on DPG-Bench would be a useful addition since the 5k evaluation set of this paper is not fully understood and only a few models have been evaluated here. Additionally, from an evaluation standpoint, even on this 5k evaluation set, VQAScore[1] might be a good option to consider, since it uses a T5-XXL model which can take long prompts, and has shown some promising results for text-to-image evaluations.

Another aspect which is missing here is that all the experiments in this paper are conducted on SD1.5 which is a relatively older model, and there have been newer models in the past 2 years (e.g. SDXL). Therefore, it would have been nicer to also have results with any of the newer, more performant models, but I can understand that this might be a bit more computationally expensive (especially if the training has to be done at 1024 resolution).

Overall, I dolike the paper, but I believe that incorporating these aspects (especially strengthening the paper with additional evaluations) could improve the paper significantly.

[1] Lin et al. "Evaluating Text-to-Visual Generation with Image-to-Text Generation", ECCV 2024

**Questions:**

I apologize in advance if I missed it, but I do not really see clear details about the training of the Denscore model. B.1 has details on the training objectives and the fact that captions are generated by LLaVA-Next, but beyond this I do not see other implementation details (dataset, other choices etc.), so it would be great if the authors could point me to this.

---

> ### Author Response · Authors · 2024-11-20
> **Rebuttal by Authors**
>
> Thank you for your positive feedback. Your insightful questions have improved our work very much. Below we respond to the comments in weaknesses (***W***) and questions (***Q***).
>
> ---
>
> ***W1: An important paper that is missed here is ELLA \[Hu et al. 2024\]. ELLA might be a valid comparison, and its adapter would likely have been a better alternative.***
>
> **Baseline:** Thank you for your advice. We have included ELLA as a new baseline in the updated version. Our method outperforms ELLA on both our original evaluation metrics and the new DPG-bench and VQAScore. Considering our training time is only 1/7 that of ELLA, this highlights the effectiveness and efficiency of our method for text alignment. The detailed results can be found in $\\textcolor{blue}{\\textrm{Table 2 and 5}}$ of the new version.
> **Adapter:** In the original paper, we selected the MLP as the adapter not for its performance but because its simplicity better emphasizes our main contribution. We acknowledge that ELLA's adapter can outperform a two-layer MLP, and we are currently working on combining our method with ELLA to create a stronger foundation model.
>
> Here is a summary of $\\textcolor{blue}{\\textrm{Table 2 and 5}}$:
>
> | Model | SD-1.5 | SD-2 | PlayG-2 | PixArt-$\\alpha$ | KanD-2.2 | ELLA-1.5 | LongSD (ours) |
> | :---- | :---- | :---- | :---- | :---- | :---- | :---- | :---- |
> | Denscore-O | 29.20 | 30.15 | 28.80 | 33.48 | 33.30 | 32.92 | **35.26** |
> | VQAscore | 84.57 | 85.61 | 85.26 | 86.96 | 86.31 | 86.85 | **87.24** |
> | DPG-Bench | 63.18 | 68.09 | 74.54 | 71.11 | 70.12 | 74.91 | **77.58** |
>
> ---
>
> ***W2: The paper ELLA also introduces DPG-Bench. Even on this 5k evaluation set, VQAScore might be a good option to consider.***
>
> Thank you again for your advice. As mentioned above, we have added DPG-bench and utilized VQAscore on our 5k test set as two new evaluation metrics for text alignment. The results show strong consistency across three evaluation metrics (Denscore-O, VQAscore, and DPG-Bench) for text alignment, and our method consistently outperforms the baselines.
>
> ---
>
> ***W3: It would have been nicer to also have results with any of the newer, more performant models (e.g. SDXL).***
>
> We have added new foundation models, SDXL. Our method also extends the maximum input token limit for SDXL and improves long-text alignment. All training and testing settings match the SD1.5 version, but with 1024 resolutions. The evaluation results on 5k test set are shown below:
>
> |  | FID | Denscore-O | Denscore | VAQscore | GPT4o |
> | :---- | :---- | :---- | :---- | :---- | :---- |
> | SDXL | 21.18 | 33.52 | 22.79 | 86.89 | 268 |
> | longSDXL | 23.88 | 37.33 | 25.33 | 87.30 | 416 |
>
> ---
>
> ***Q1: I do not see other implementation details (dataset, other choices etc.) regarding the training of the Denscore model.***
>
> We apologize for this. The Denscore training setup is briefly described at the end of the training part in Section 5.1. We maintain consistency with Pickscore across nearly all settings, except for the training objectives and datasets. Specifically, we train our Denscore using Pickscore’s [GitHub repository](https://github.com/yuvalkirstain/PickScore), making only modifications to the dataset loading and loss function code.

---

> > ### Comment · Reviewer_hkae · 2024-11-25
> >
> > I thank the authors for taking the time to provide the clarifications and updating the paper. I think the additions (ELLA, DPG-Bench, VQAScore, SDXL, and providing implementation details) makes the paper much stronger and I happy to recommend acceptance of the paper.
> > A minor observation: Tab. 7 has 2 columns named VQAScore, I believe the first one should be Denscore? In general, it would be a good idea to carefully check the paper for typos once especially given that there are a lot of additions to the paper.

---

> > > ### Author Response · Authors · 2024-11-26
> > > **Thank you for your support**
> > >
> > > Thank you for your support. We appreciate your detailed feedback and valuable suggestions, which have been instrumental in improving our work. We have corrected the typos in Table 7 and are conducting a thorough review of the paper, including refining the overall flow to better integrate the additions made during the rebuttal. Thank you once again for your valuable input!

---

### Official Review · Reviewer_9HBG · 2024-11-03

**Soundness:** 2
**Presentation:** 3
**Contribution:** 2
**Rating:** 3
**Confidence:** 4

**Summary:**

This paper proposes a novel method to improve text-to-image (T2I) diffusion models in handling long text inputs. Due to the input length limitations of existing encoders like CLIP, it becomes challenging to accurately align generated images with long texts. To address this issue, the authors propose a segment-level encoding strategy, which divides long texts into segments and encodes them separately, combined with a decomposed preference optimization method to reduce overfitting and enhance alignment. Experimental results show that the fine-tuned model surpasses several existing foundation models in long-text alignment, demonstrating significant improvements in handling long text inputs.

**Strengths:**

[1] It introduces a segment-level encoding strategy that effectively handles long text inputs by dividing and separately encoding segments, overcoming traditional model input limitations and enhancing text-to-image alignment.
[2] The preference model is innovatively decomposed into text-relevant and text-irrelevant components, with a reweighting strategy to reduce overfitting and improve alignment precision.
[3] The paper conducts extensive experiments, demonstrating significant improvements in long-text alignment over existing models like PixArt-α and Kandinsky v2.2, proving the method's effectiveness for complex text generation tasks.

**Weaknesses:**

[1] The paper proposes a segment-level encoding strategy to handle long texts but does not thoroughly validate the performance of this strategy under different text length conditions. For very short or very long texts, can the segment-level encoding still maintain the same alignment effectiveness? The lack of fine-grained comparative experiments makes it difficult to adequately demonstrate the applicability of segment-level encoding across a wide range of text lengths.
[2] The paper proposes a reweighting strategy to address overfitting, but lacks detailed experimental data to demonstrate its effectiveness, failing to adequately prove its specific impact on reducing overfitting.
[3] The segment-level encoding and preference optimization strategies proposed in this paper show excellent performance in the experiments, but lack an analysis of the method's limitations. It would be beneficial to discuss whether these segment-level encoding methods might lose part of their alignment effectiveness when dealing with texts that have complex contextual dependencies or require strong semantic understanding.

**Questions:**

[1] Does your proposed segment-level encoding strategy demonstrate significant effectiveness for texts of varying lengths? Specifically, how does the model perform with very short texts (fewer than 10 words) or very long texts (over 500 words)? Could you provide additional experiments to show comparative results under different text length conditions to verify the generalizability of the segment-level encoding strategy?
[2] You mentioned using a reweighting strategy to mitigate the model's overfitting issue, but the description of this process in the paper is rather brief. Could you provide detailed steps or pseudocode to explain the implementation of this strategy? Additionally, does this method have any quantitative results to demonstrate its effectiveness in reducing overfitting in specific scenarios? Could you include comparative data from the experiments to validate the impact of this strategy?
[3] How were the 5k images in the test set specifically selected from datasets like SAM and COCO2017?
[4] Could you briefly explain the selection of models like CLIP-H and HPSv2 in the experimental section of Chapter 5, as well as the chosen evaluation metrics?

**Details Of Ethics Concerns:**

N.A.

---

> ### Author Response · Authors · 2024-11-20
> **Rebuttal by Authors [1/2]**
>
> Thank you for your constructive feedback. Your insightful questions have improved our work very much. Below we respond to the comments in weaknesses (***W***) and questions (***Q***).
>
> ---
>
> ***W1 & Q1: The performance of the segment-level encoding strategy under different text length conditions.***
>
> In $\\textcolor{blue}{\\textrm{Table 6}}$ of our revised paper, we assess the generation results for prompts of lengths about 15, 60, 120, 240, and 500 tokens using Denscore-O and VQAscore \[1\]. For more details on the dataset construction and evaluation process, please see Appendix C.3 of the new version. The evaluation results in $\\textcolor{blue}{\\textrm{Table 6}}$ demonstrate that our method consistently outperforms current baselines.
>
> Here is a summary of the **VQAscore results** in $\\textcolor{blue}{\\textrm{Table 6}}$ in Appendix C.3:
>
> | Token | SD-1.5 | SD-2 | PlayG-2 | PixArt-$\\alpha$ | KanD-2.2 | ELLA-1.5 | LongSD (ours) |
> | :---- | :---- | :---- | :---- | :---- | :---- | :---- | :---- |
> | 15 | 88.32 | 90.27 | 88.32 | 91.12 | 91.88 | 90.28 | **92.52** |
> | 120 | 83.63 | 85.33 | 84.78 | 86.81 | 86.02 | 86.93 | **87.49** |
> | 500 | 81.14 | 82.69 | 80.42 | 84.79 | 84.94 | 85.84 | **87.24** |
>
> \[1\] Lin Z, Pathak D, Li B, et al. Evaluating text-to-visual generation with image-to-text generation\[C\]//European Conference on Computer Vision. Springer, Cham, 2025: 366-384.
>
> ---
>
> ***W2 & Q2: Does the reweighting strategy have any quantitative results to demonstrate its effectiveness in reducing overfitting? Please provide detailed steps or pseudocode of it.***
>
> The pseudocode of the reweighting strategy is
> ```python
> # Calculate the text-unrelated component $V$
> for image, text in dataset:
>     text_emb_list.append(CLIP(text)  / ||CLIP(text)||)
> common_text_emb = mean(text_emb_list) / ||mean(text_emb_list)||
>
> # Calculate the reweighted loss
> image_emb = CLIP(image) / ||CLIP(image)||
> text_emb = CLIP(text)  / ||CLIP(text)||
> # Equation 8
> # Ratio controls the reweighted proportion
> text_emb_reweight = text_emb - (1 - ratio) * (text_emb * common_text_emb) * common_text_emb
> loss = text_emb_reweight * image_emb
> ```
> The quantitative results for reducing overfitting are presented in $\\textcolor{blue}{\\textrm{Figure 5}}$. When the ratio approaches 1, the Denscore decreases while the FID (FID evaluates the distribution distance between the dataset and generated images) increases, indicating generation overfitting to the Denscore and a departure from the correct image distribution. At a ratio of 0.3, we can maintain a relatively stable FID, suggesting that generated images stay within the desired distribution. Visual results are available in $\\textcolor{blue}{\\textrm{Figures 17}}$ of the new version. At a ratio of 1.0, there is clear evidence of overfitting, as all images show similar patterns; whereas at a ratio of 0.3, the images align best with human preferences without obvious overfitting.
> In addition to the experimental results, our paper identifies the main cause of overfitting: when training diffusion models with CLIP-based preference models, the model tends to optimize towards the text-unrelated component $\\mathbf{V}$, resulting in generated images that appear similar regardless of the text input. Based on this finding, we choose to reweight the item $\\mathbf{V}$ during training.
>
> ---
>
> ***W3: Discuss the alignment effectiveness when dealing with texts that have complex contextual dependencies or require strong semantic understanding.***
>
> We would like to clarify that aligning long texts and complex texts are two distinct issues. A single-sentence prompt can also have intricate dependencies, but our paper focuses more on the length aspect. To demonstrate the effectiveness of our current methods in handling complex dependencies and semantic understanding we use DPG-Bench \[2\], which includes test prompts for various categories such as entity, attribute, relation, and count. The new results are shown in $\\textcolor{blue}{\\textrm{Table 5}}$ of Appendix C.3 of the new version. Compared to the baselines, we have also made obvious improvements. In addition, we agree that complex dependencies and semantic understanding are challenging tasks and remain far from being fully resolved. We have also included these points in the limitations section of the new version, with changes highlighted in blue for clarity.
>
> Here is a summary of $\\textcolor{blue}{\\textrm{Table 5}}$ in Appendix C.3:
>
> | Model | SD-2 | PlayG-2 | PixArt-$\\alpha$ | KanD-2.2 | SD-1.5 | ELLA-1.5 | LongSD (ours) |
> | :---- | :---- | :---- | :---- | :---- | :---- | :---- | :---- |
> | DPG-Bench | 68.09 | 74.54 | 71.11 | 70.12 | 63.18 | 74.91 | **77.58** |
>
> \[2\] Hu X, Wang R, Fang Y, et al. Ella: Equip diffusion models with llm for enhanced semantic alignment\[J\]. arXiv preprint arXiv:2403.05135, 2024\.

---

> ### Author Response · Authors · 2024-11-20
> **Rebuttal by Authors [2/2]**
>
> ***Q3: How were the 5k images in the test set specifically selected from datasets like SAM and COCO2017?***
>
> We first collect about 2 million images from open datasets, including 500k from SAM, 100k from COCO2017, 500k from LLaVA (a subset of the LAION/CC/SBU dataset), and 1 million from JourneyDB. After obtaining this new dataset, we randomly select 5k images from it as the test set without any human intervention. The 5k test set is not used in any training stage. More information about our dataset can be found in Section 5.1.
>
> ---
>
> ***Q4: Briefly explain the selection of CLIP-H and HPSv2, as well as the chosen evaluation metrics?***
>
> Thank you for your advice. We have added more information about these methods in Section 5.1 and 5.2 of the new version. The explanations are as follows:
> * CLIP-H and HPSv2: To demonstrate that our analysis of the CLIP-based preference models is generally applicable, we compare four different CLIP-based models: the pretrained CLIP, the single-value preference models Pickscore and HPSv2, as well as our segment-level preference model, Denscore.
> * FID: FID evaluates the distribution distance between the dataset and generated images.
> * Denscore: Denscore assesses human preference for generated images, while Denscore-O and VQAscore focuses on the text alignment of those images.
> * VQAscore: VQAscore \[1\] also focuses on the text alignment of generated images.
> * DPG-bench: DPG-bench \[2\] is a general benchmark that includes test prompts for categories like entity, attribute, relation, and count.

---

> ### Author Response · Authors · 2024-11-25
> **Looking forward to further feedback**
>
> Dear Reviewer 9HBG,
>
> Thank you once again for your constructive feedback. We would like to kindly remind you that we have included additional experiments to:
>
> - Validate the performance of our approach under varying text-length conditions.
> - Assess alignment effectiveness when handling complex texts.
> - Provide detailed evidence of the effectiveness of the reweighting strategy in addressing overfitting.
>
> We have also clarified the construction of the test set, the selection of models (CLIP-H and HPSv2), and the evaluation metrics used.
>
> ---
>
> As the discussion period is coming to a close in two days, we look forward to your response and would be happy to address any further comments or questions you may have.
>
> Best,
>
> The Authors

---

> > ### Comment · Reviewer_9HBG · 2024-11-26
> >
> > Thank you for the response. Two quick questions:
> > 1. Please elaborate a case of the input data that is a long but not complex (for better illustration, please also include an example of a long and complex input). I am not convinced by the claim that a long text may be purely long, rather that having more text to deliver more meaningful semantic information.
> >
> > 2. Viewing that the LLMs may well-handle long text to some extent. Please brief a case where the proposed algorithm may win.

---

> > > ### Author Response · Authors · 2024-11-26
> > > **Thank you for your feedback**
> > >
> > > Thank you for your feedback\! Below, we address the two new questions (***Q***).
> > >
> > > ---
> > >
> > > ***Q1: Please elaborate a case of the input data that is long but not complex (for better illustration, please also include an example of a long and complex input).***
> > >
> > > We appreciate the opportunity to provide further clarification. The subtle difference between "long" and "complex" input texts lies in whether the text contains redundant information that can often be inferred from other parts of the text, versus containing rich details related to quantities, positions, relative relationships, or intricate dependencies.
> > >
> > > An example of a long but not very complex input text could be:
> > >
> > > > *In this captivating photograph, a giant panda sits serenely amidst a lush green backdrop of bamboo, creating a striking contrast with its surroundings. The panda’s distinctive black-and-white fur stands out beautifully against the vibrant greenery, emphasizing its unique appearance. As it rests peacefully, the serene expression on its face reflects the tranquility of its environment. Surrounded by towering bamboo, the scene encapsulates the essence of this beloved species, showcasing the giant panda in its natural habitat, where its striking black-and-white fur harmonizes with the lush green backdrop.*
> > >
> > > In this example, descriptive elements like "black-and-white fur" or "lush green backdrop of bamboo" could be inferred from the general context, as they repeat or reinforce ideas present elsewhere in the text.
> > >
> > > An example of a shorter but more complex input text could be:
> > >
> > > > *The image shows five interconnected ecosystems, each containing 12 plants and 8 animals.*
> > >
> > > While this text is shorter, it is more complex due to its intricate relationships and specific numerical details, which require precise alignment of ecosystems, plants, and animals.
> > >
> > > While the examples illustrate some distinctions between "long" and "complex," we acknowledge that "complexity" is inherently abstract and hard to quantify. Moreover, we agree that longer texts often correlate with greater complexity due to the potential for richer semantic information. In response to this, our results on DPG-Bench ($\\textcolor{blue}{\\textrm{Table 5}}$ in Appendix C.3) demonstrate that our method effectively handles input texts even when they involve intricate dependencies and complex semantics.
> > >
> > > ---
> > >
> > > ***Q2: Viewing that the LLMs may well-handle long text to some extent. Please brief a case where the proposed algorithm may win.***
> > >
> > > In our paper, we observe that combining CLIP with LLMs like T5 as encoders is more effective than using T5 alone, as shown in $\\textcolor{blue}{\\textrm{Figure 4}}$. This effectiveness is probably due to contrastive pre-training encoders (e.g., CLIP) being specifically designed for text-image alignment, which potentially enhances the correspondence between text representations and generated images. Additional details are available in Section 5.3 of our paper.
> > >
> > > However, existing pretrained CLIP models have a maximum token limit. Our segment encoding enables us to extend this limit, allowing it to process much longer inputs (e.g., 250 to 500 tokens). Moreover, such long prompts can often be divided into several relatively independent segments, making our segment encoding both practical and logical. Supporting experiments about long inputs can be found in $\\textrm{\\color{blue}Table 6}$ of our paper.
> > >
> > > Additionally, our whole method includes both segment encoding and decomposed preference optimization. Decomposed preference optimization is a training strategy that is beneficial for long-text alignment regardless of whether LLMs or CLIP are used as encoders. For example, the experiments of longSD in $\\textrm{\\color{blue}Table 2}$ use both T5 and CLIP as encoders; the fine-tuned version with decomposed preference optimization (S+R) performs significantly better than the version (S) without it.

---

> > > > ### Comment · Reviewer_9HBG · 2024-11-26
> > > >
> > > > Received with thanks!

---

> > > > > ### Author Response · Authors · 2024-11-27
> > > > > **Thank you for your feedback**
> > > > >
> > > > > Thank you for your prompt feedback! We appreciate your time and thoughtful comments. Since a rating of 3 is still considered negative, we would be grateful if you could share any remaining concerns you might have. If you find our responses satisfactory, we kindly ask if you might consider revising your rating based on your updated evaluation of our work.
> > > > >
> > > > > Thank you again for your valuable input.

---

> > > > > > ### Comment · Reviewer_9HBG · 2024-12-03
> > > > > >
> > > > > > Considering the responses and the concerns of other reviewers, I think the proposed solution does not well-answer the theoretical issues raised by long text, and its main contribution is to make the more become more trainable under the input of long text. I would like to keep my rating to make it a boarderline, and it is fine for me if the paper is accepted considering its contribution on the model.

---

> > > > > > > ### Author Response · Authors · 2024-12-03
> > > > > > > **Thank you for your feedback**
> > > > > > >
> > > > > > > Dear Reviewer 9HBG,
> > > > > > >
> > > > > > > Thank you for your feedback, and we appreciate your openness to the acceptance of our work. While we understand that the remaining time for responses is very limited, we would like to kindly ask for clarification on which specific concerns from other reviewers and what specific theoretical issues related to long text you are referring to. Since authors still have over a day to respond, we hope to have the opportunity to provide further clarification to address your concerns and improve your evaluation of our work during the Reviewer/AC discussions.
> > > > > > >
> > > > > > > \
> > > > > > > Regarding your statement:
> > > > > > > > I would like to keep my rating to make it a boarderline, and it is fine for me if the paper is accepted considering its contribution on the model.
> > > > > > >
> > > > > > > We understand that you may have reservations about our work and consider it borderline. However, we would like to note that a rating of **3** is not typically intended for borderline papers. The rating guidelines suggest using a score of **5** for "*marginally below the acceptance threshold*" and **6** for "*marginally above the acceptance threshold*." **Balancing the overall rating rather than rating based on an individual reviewer's assessment might not align with the expectations of the program committee.**
> > > > > > >
> > > > > > > \
> > > > > > > Nevertheless, we are truly grateful for your constructive review and thoughtful feedback. Please let us know if there are any remaining concerns we can address.
> > > > > > >
> > > > > > > Best regards,\
> > > > > > > The Authors

---

### Official Review · Reviewer_GqtG · 2024-11-04

**Soundness:** 3
**Presentation:** 3
**Contribution:** 3
**Rating:** 8
**Confidence:** 4

**Summary:**

This paper presents a novel approach to enhance the alignment between long text descriptions and generated images in text-to-image diffusion models, introducing segment-level encoding to overcome input length limitations and decomposed preference optimization to mitigate overfitting and improve text-relevant alignment during fine-tuning.

**Strengths:**

1. The motivation for using preference models is well-founded, and the paper is well-written.
2. It is interesting to identify two distinct focuses within preference models, and the analysis provided is both reasonable and thorough.

**Weaknesses:**

weakness
1. I am unsure why multiple <sot> tokens are retained; regarding the retention or removal of tokens, a more detailed explanation or analysis is needed, as it currently leaves me confused.
2.After reweighting, whether there will be a noticeable difference in the aesthetic quality of the generated results (due to text-irrelevant components) remains unclear. For Appendix B.1, it would be beneficial to provide some visualizations of the outcomes from the two loss functions.
3. Segmenting to leverage CLIP's alignment effect is an intuitive innovation, but does this become irrelevant in light of the development of Vision-Language Models (VLMs)? Can the current innovation still contribute to VLMs?
4. On line 363, it mentions mitigating the risk of overfitting to Denscore. Could you clarify where the potential source of this overfitting lies?

**Questions:**

see Weaknesses

---

> ### Author Response · Authors · 2024-11-20
> **Rebuttal by Authors**
>
> Thank you for your positive feedback. Your insightful questions have improved our work very much. Below we respond to the comments in weaknesses (***W***).
>
> ---
>
> ***W1: Why are multiple \<sot\> tokens retained?***
>
> Thank you for raising this question. In the revised paper, we have added an ablation study ($\\textcolor{blue}{\\textrm{Figure 10 and 11}}$ in Appendix A) to explore three scenarios: (1) removing all \<sot\> tokens, (2) retaining a single \<sot\> token, and (3) retaining all \<sot\> tokens. Our findings indicate that removing all \<sot\> tokens leads to a significant performance drop, while keeping one or more \<sot\> tokens yields comparable results. This analysis is elaborated in Appendix A, with changes highlighted in blue for clarity.
>
> ---
>
> ***W1.2: The difference in the aesthetic quality of the generated results after reweighting.***
>
> In the revised paper, we have provided multiple generation images ($\\textcolor{blue}{\\textrm{Figure 17}}$ in Appendix C.2) at different reweight ratios while keeping all other settings the same to illustrate differences in aesthetic quality. A ratio of 1 indicates the original preference loss, resulting in significant overfitting, where all images exhibit similar patterns regardless of the inputs. A ratio of 0 implies that the loss only considers the text-relevant part, leading to low image quality that does not align with human preferences. We observe a ratio of 0.3 yields the best visual quality among these options.
>
> ---
>
> ***W1.3: Some visualizations of the outcomes from the two loss functions of Denscore.***
>
> For long-text alignment, the two losses are similar; however, for aesthetics, Equation 10 eliminates the influence of the first item on the text-irrelevant part $V$, allowing it to focus more on aesthetic aspects. In $\\textcolor{blue}{\\textrm{Figure 13}}$ of Appendix B.1 in the revised paper, we present updated visual results for retrievals with the highest scores using the text-irrelevant part $V$ under two Denscore losses. We find that retrieval results from the text-irrelevant part $V$ trained with Equation 10 align more closely with human aesthetic preferences.
>
> ---
>
> ***W2: Can the current innovation still contribute to VLMs?***
>
> An important direction for VLM is to enable it to understand and generate visual input simultaneously \[1,2\]. Our methods could also be potentially used to train VLMs to generate images similarly to T2I diffusion models. Furthermore, in CLIP, there is a symmetrical relationship between image and text. By reversing their roles—using the image as input and generating text as output—we could also enhance VLM’s image captioning capabilities \[3\], presenting another interesting potential benefit.
>
> \[1\] Team C. Chameleon: Mixed-modal early-fusion foundation models\[J\]. arXiv preprint arXiv:2405.09818, 2024\.
> \[2\] Zhou C, Yu L, Babu A, et al. Transfusion: Predict the next token and diffuse images with one multi-modal model\[J\]. arXiv preprint arXiv:2408.11039, 2024\.
> \[3\] Liu H, Li C, Wu Q, et al. Visual instruction tuning\[J\]. Advances in neural information processing systems, 2024, 36\.
>
> ---
>
> ***W3: Could you clarify where the potential source of this overfitting lies?***
>
> Previous work \[4\] has observed the overfitting problem in preference optimization of diffusion models, while our paper identifies its main cause and solves it. We find that when using CLIP-based preference models to train diffusion models, the training loss can be divided into a text-related component $\\mathcal{C}\_{P}^{\\bot}(p) \* \\mathcal{C}\_{X}$ and a text-unrelated component $\\mathbf{V} \* \\mathcal{C}\_{X}$. The text-unrelated component makes up a large part of the entire loss, and it is relatively easy for the model to learn since $\\mathbf{V}$ remains stable. As a result, the model tends to optimize towards $\\mathbf{V}$ regardless of the text input, leading all generated images to look similar. More information can be found in Section 4.2, and visual results can be found in $\\textcolor{blue}{\\textrm{Figures 6 and 17}}$ in the revised paper.
>
> \[4\] Wu X, Hao Y, Zhang M, et al. Deep Reward Supervisions for Tuning Text-to-Image Diffusion Models\[J\]. arXiv preprint arXiv:2405.00760, 2024\.

---

> > ### Comment · Reviewer_GqtG · 2024-11-28
> > **Response**
> >
> > The author addresses my concerns, and based on the overall quality, I will maintain the final rate

---

> > > ### Author Response · Authors · 2024-11-28
> > > **Thank you for your support**
> > >
> > > We appreciate your kind support! In our final revision, we will further improve the paper by incorporating the valuable insights gained from the rebuttal discussions. Thank you again!

---

### Author Response · Authors · 2024-11-20
**Summary of Paper Revision**

We thank all reviewers for their constructive feedback, and we have responded to each reviewer individually. We have also uploaded a **Paper Revision** including additional results and illustrations:

* $\\textrm{\\color{blue}Figure 11}$: Generation results of segment encoding using different numbers of the \<sot\> token and segmentation strategies. (For reviewers GqtG and Hfc6.)
* $\\textrm{\\color{blue}Figure 12}$: Visual results of attention maps before and after training with our methods. (For reviewer Hfc6.)
* $\\textrm{\\color{blue}Figure 13}$: Retrieval results using the text-irrelevant part $\\mathbf{V}$ from Denscore models trained with different loss functions. (For reviewer GqtG.)
* $\\textrm{\\color{blue}Figure 17}$: Visual results using reweighting strategies with various ratios. (For reviewers GqtG and 9HBG.)
* $\\textrm{\\color{blue}Table 2}$: The new version includes an updated baseline ELLA \[1\] and a new evaluation metric, VQAscore \[2\]. (For reviewer hkae.)
* $\\textrm{\\color{blue}Table 5}$: Evaluation results using DPG-Bench \[1\] for a wider range of prompts. (For reviewers 9HBG, hkae and Hfc6.)
* $\\textrm{\\color{blue}Table 6}$: Evaluation results using prompts of varying lengths: 15, 60, 120, 240, and 500 tokens. (For reviewers 9HBG and Hfc6.)

\[1\] Hu X, Wang R, Fang Y, et al. Ella: Equip diffusion models with llm for enhanced semantic alignment\[J\]. arXiv preprint arXiv:2403.05135, 2024\.
\[2\] Lin Z, Pathak D, Li B, et al. Evaluating text-to-visual generation with image-to-text generation\[C\]//European Conference on Computer Vision. Springer, Cham, 2025: 366-384.

---

### Author Response · Authors · 2024-11-23
**Looking forward to further feedback**

Dear Reviewers,

Thank you again for your valuable comments and suggestions, which are really helpful for us. We have posted responses to the concerns raised in your reviews and included additional experiment results.

We totally understand that this is quite a busy period, so we deeply appreciate it if you could take some time to return further feedback on whether our responses solve your concerns. If there are any other comments, we will try our best to address them.

Best regards,

The Authors

---

### Meta-Review · Area_Chair_t8Ur · 2024-12-17

**Metareview:**

This paper proposes a novel text encoder to deal with long conditioning texts in text-to-image diffusion models. Long texts are divided into segments and processed separately. The authors finetune a stable diffusion model based on this encoding, using a CLIP-based preference optimization method.
Strengths mentioned in the reviews include: well written paper, good review of related work, good analysis of preference model leading to decomposition, and significant improvements over prior work.
Weaknesses included: missing some baseline comparisons, experiments only used the SD1.5 as generative foundation model, missing evaluations as function of prompt length, there were also suggestions to include DPG-bench and VQAScore for evaluation.

**Additional Comments On Reviewer Discussion:**

Based on the initial reviews the authors submitted a rebuttal and revised manuscript to address the points raised by the reviewers. The rebuttal addresses most of the concerns of the reviewers, which in the light of the response recommended accepting the paper (3x), the single reviewer that was not quite satisfied with the author response to their questions indicated they were not arguing against acceptance of the paper despite their rating. The AC therefore follows the majority recommendation of accepting the paper.

---

### Decision · Program_Chairs · 2025-01-22

Accept (Poster)